# What Secrets Do Your Manifolds Hold? Understanding the Local Geometry of Generative Models

**Ahmed Imtiaz Humayun**[1,2]**, Ibtihel Amara**[1,3]**, Cristina Vasconcelos**[4]**,**
**Deepak Ramachandran**[4]**, Candice Schumann**[4]**, Junfeng He**[1]**, Katherine Heller**[1]**,**
**Golnoosh Farnadi**[1]**, Negar Rostamzadeh**[1]**, Mohammad Havaei**[1]
[1]Google Research, [2]Rice University, [3]McGill University, [4]Google Deepmind
imtiaz@rice.edu, mhavaei@google.com

## Abstract

Deep Generative Models are frequently used to learn continuous representations of complex data distributions by training on a finite number of samples. For any generative model, including pre-trained foundation models with Diffusion or Transformer architectures, generation performance can significantly vary across the learned data manifold. In this paper, we study the local geometry of the learned manifold and its relationship to generation outcomes for a wide range of generative models, including DDPM, Diffusion Transformer (DiT), and Stable Diffusion 1.4. Building on the theory of continuous piecewise-linear (CPWL) generators, we characterize the local geometry in terms of three geometric descriptors - scaling ($\psi$), rank ($\nu$), and complexity/un-smoothness ($\delta$). We provide quantitative and qualitative evidence showing that for a given latent vector, the local descriptors are indicative of post-generation aesthetics, generation diversity, and memorization by the generative model. Finally, we demonstrate that by training a reward model on the *local scaling* for Stable Diffusion, we can self-improve both generation aesthetics and diversity using geometry sensitive guidance during denoising. Website: imtiazhumayun.github.io/generative_geometry.

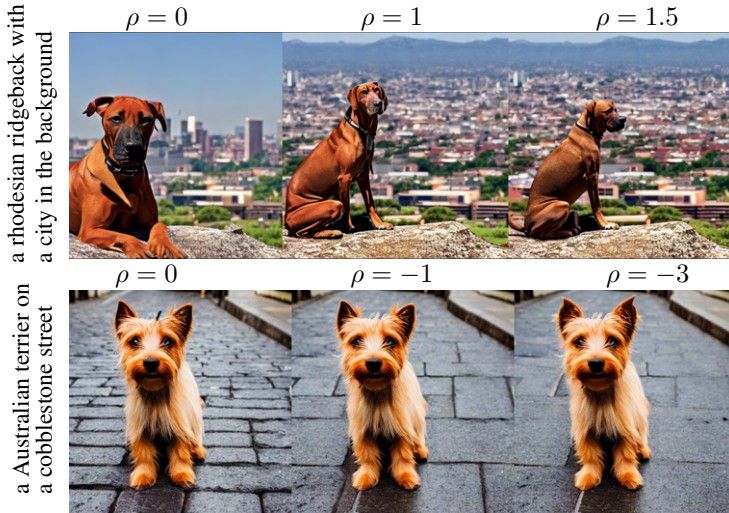

Figure 1: **Geometry sensitive diffusion guidance.** During the reverse diffusion process, increasing **(top-row)** or decreasing **(bottom-row)** the local scaling, i.e., volume dilation, of denoising trajectories using geometry reward based guidance with strength $\rho$, results in increased **(top-row)** or decreased **(bottom-row)** visual complexity. Increasing local scaling results in more background elements coming into view and a decrease in the focus on the subject, vice-versa when local scaling is decreased. Here we use a reward model trained on the *local scaling* computed for $800K$ Stable Diffusion (Rombach et al., 2021) latents. More examples in appendix fig. 22, fig. 23, fig. 24.

## 1 INTRODUCTION

In recent years, deep generative models have emerged as a powerful tool in machine learning, capable of synthesizing realistic data across diverse domains (Karras et al., 2019; 2020; Rombach et al., 2021). However, the performance of such generative models may vary based on the latent vector or (text-)conditioning used for generation. Recent studies have demonstrated that models like Stable Diffusion can exhibit generation biases in terms of reduced generation fidelity or diversity, when certain demographic groups are mentioned in the prompts Zhao et al. (2018); Luccioni et al. (2023). Especially for *1)* models trained with large heterogeneous data distributions, and for *2)* generative models pre-trained on unknown training distributions, the aforementioned observations, i.e., latent/prompt specific behaviors can become hard to interpret or reason. Since generative models are essentially approximating a *generative function* mapping a latent space to the data manifold, one way to explain downstream performance could be in terms of the characteristics of the approximated function that can be measured via the model internals, i.e., activations and weights. In this regard, we pose the following research question:

**Research Question.** *For any deep generative model pre-trained on an arbitrary dataset, how is the local geometry of the generator function related to downstream generation?*

The theory of continuous piecewise-linear generators Balestriero et al. (2020) suggests that a large class of generative models can be considered continuous piecewise linear (CPWL) operators, implying that such generative models can be fully characterized in terms of their weights and architecture. We consider it the framework of choice to find answers to the aforementioned question and propose using three *local geometric descriptors* to quantify local characteristics of any pre-trained generative model:

- **Local rank** ($\nu$), that characterizes the local dimensionality of the learned manifold.

- **Local scaling** ($\psi$), that characterizes the local change of volume by the input output mapping of the generative model.

- **Local complexity** ($\delta$), that approximates the *un-smoothness* of the generative model in terms of second order changes in the input-output mapping.

Geometric descriptors such as local scaling, complexity or rank, have previously been used to measure function complexity of Deep Neural Networks (DNN) (Hanin & Rolnick, 2019) and DNN expressivity (Poole et al., 2016; Raghu et al., 2017), to evaluate the quality of representations learned with a self-supervised objective (Garrido et al., 2023), for interpretability and visualization of DNNs (Humayun et al., 2023), to understand the learning dynamics in reinforcement learning (Cohan et al., 2022), to explain grokking, i.e., delayed generalization and robustness in classifiers (Humayun et al., 2024), sampling of GAN based generative models (Humayun et al., 2021; 2022b), and maximum likelihood inference in the latent space (Kuhnel et al., 2018). We provide a more extensive discussion on related works in appendix B.

**Our contributions.** In this paper, through rigorous experiments on large image generative models, we establish correlations between the local geometry of the generative function and the aesthetics of the generation, the diversity of the generation, and the memorization in the generated samples. We demonstrate how these qualities manifest differently for different sub-populations of the learned distribution. We also show that the geometry of the data manifold is heavily influenced by the training data, which enables applications in out-of-distribution detection and reward modeling to control the output distribution. Our empirical results lead to the following conclusions.

- **C1.** We present the first large-scale analysis of the local geometry of foundational text-to-image latent diffusion models and establish correlations between local geometric descriptors and downstream aesthetic quality, diversity, and memorization (Sec 4.).

- **C2.** For small diffusion models and foundational image generative models, we show that the local geometry on the generative model manifold is distinct from the off-manifold geometry and can help distinguish the domain of a generative model (Sec. 3).

- **C3.** By training an auxiliary model on precomputed local geometric descriptors of Stable Diffusion, we present a novel framework for reward guidance on a diffusion model to increase/decrease sampling diversity or control aesthetic qualities (Sec 5).

## 2    LOCAL DESCRIPTORS OF GENERATIVE MODEL MANIFOLDS

We start by introducing the geometric descriptors we will use in our study and provide insight into what aspect of the generative model manifold geometry each of the descriptors quantify.

### 2.1    CONTINUOUS PIECEWISE-LINEAR GENERATIVE MODELS

Consider a generative network $\mathcal{G}$, which can be the decoder of a Variational Autoencoder (VAE) (Kingma & Welling, 2013), the generator of a Generative Adversarial Network (GAN) (Goodfellow et al., 2014), or an unrolled denoising diffusion implicit model (DDIM) (Song et al., 2020). Suppose, $\mathcal{G} : \mathbb{R}^E \rightarrow \mathbb{R}^D$ is a deep neural network with $L$ layers, input space dimensionality $E$ and output space dimensionality $D$. For any such generator, if the layers comprise affine operations such as convolutions, skip-connections, or max/avg-pooling, and the non-linearities are continuous piecewise-linear (CPWL) such as leaky-ReLU Xu (2015), ReLU, or periodic triangle, then the generator is a continuous piecewise-linear operator (Balestriero & Baraniuk, 2018a; Humayun et al., 2023). This implies that the $\mathcal{G} : \mathbb{R}^E \rightarrow \mathbb{R}^D$ mapping can be expressed in terms of a subdivision of the input space into linear regions $\Omega$ with each region $\omega$ from the input domain being mapped to the output via an affine operation. The continuous data manifold or image of the generator $\text{Im}(\mathcal{G})$ can be written as the union of sets:

$$\text{Im}(\mathcal{G}) = \bigcup_{\forall \omega \in \Omega} \{ \boldsymbol{A}_\omega z + \boldsymbol{b}_\omega \forall z \in \omega \}, \tag{1}$$

where, $\Omega$ is the partition of the latent space $\mathbb{R}^E$ into continuous piecewise-linear regions, $\boldsymbol{A}_\omega$ and $\boldsymbol{b}_\omega$ are the slope and offset parameters of the affine mapping from latent space vectors $z \in \omega$ to the data manifold. For the class of continuous piecewise-linear (CPWL) neural network based generative models, $\Omega$, $\boldsymbol{A}_\omega$, and $\boldsymbol{b}_\omega$ are functions of the neurons/parameters of the network. For a generator with $L$ layers, $\boldsymbol{A}_\omega$ and $\boldsymbol{b}_\omega$ can be expressed in closed-form in terms of the weights and the region-wise activation pattern of neurons for each layer. We refer the readers to Lemma 1 of (Humayun et al., 2023) for details.

To help build intuition, without loss of generality lets consider a CPWL toy generator that is trained on a handcrafted task where the target function $\tilde{f} : \mathbb{R}^2 \rightarrow \mathbb{R}^3$ is a mapping between $\mathbb{R}^2$ and a mixture of five gaussian functions. Since the learned function is a continuous piecewise-affine spline operator, we use SplineCAM (Humayun et al., 2023) to analytically compute the function learned by the generator and visualize the learned manifold, as well as the input space piecewise-linear partition learned by the generator in Fig. 2 middle-left and left. Each convex region $\omega$ bounded by the black lines, is mapped to $\text{Im}(\mathcal{G})$ via per region parameters as described in Equation 1. The input-output mapping operation by the generator is affine region-wise, therefore any given input space region can be *scaled, rotated or translated* with a continuity constraint between regions, while going from the input to the output. For CPWL generators there are three characteristics of the learned manifold that can be studied: *i) the affine scaling induced per region, ii) the number of dimensions that are retained after scaling, i.e., local dimensionality of the learned manifold, and iii) the local smoothness of the CPWL partition.* We now introduce local descriptors that can be used to characterize these quantities.

### 2.1.1    LOCAL SCALING, $\psi$

We first introduce local scaling as a target descriptor to be used in our study that measures the local scaling performed on a region $\omega$ by a CPWL generator.

**Definition 1.**    For a CPWL manifold produced by generator $\mathcal{G}$, the *local scaling* $\psi_\omega$ is constant within each region $\omega$, and measures the log-scaling of the volume induced by the affine slope $\boldsymbol{A}_\omega$ for all latents $\boldsymbol{z} \in \omega$. Local scaling for $\omega$ is expressed as

$$\psi_\omega = \log(\sqrt{\det(\boldsymbol{A}_\omega^T \boldsymbol{A}_\omega)}) = \sum_i^k \log(\sigma_i) \mathbb{1}_{\{\sigma_i \neq 0\}}, \tag{2}$$

where, $\{\sigma_i\}_{i=0}^{i=k}$, are the non-zero singular values of $\boldsymbol{A}_\omega$.

Refering back to the example in Fig. 2, each region on the CPWL manifold (middle-left) and in the input space (left) is colored by $\psi_\omega$, with darker shades indicating higher $\psi_\omega$. Suppose $\mathcal{G}$ has a uniform latent distribution, meaning every region $\omega$ has a uniform probability density in the latent space.

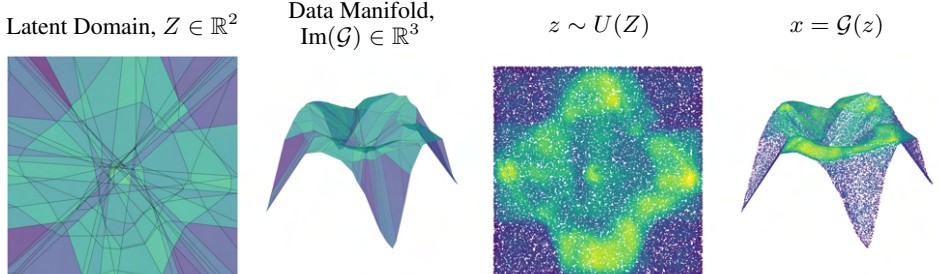

Figure 2: **The geometry of a continuous piecewise-linear toy generator.** For a CPWL generator $\mathcal{G} : \mathbb{R}^2 \to \mathbb{R}^3$, we provide analytically computed visualization of the input space partition, i.e., arrangement of linear regions (left) and learned CPWL manifold (middle-left). Each piece for this example, is colored by the piecewise-constant scaling induced by $\mathcal{G}$. Uniform samples from the latent domain (middle-right) and generated samples (right) are also presented, colored by the density of the output distribution estimated using a gaussian kernel density estimator in $\mathbb{R}^3$. We see that for any sample $z \in \omega$, the estimated density ($\uparrow$ green) is inversely proportional to the scaling ($\downarrow$ green) for region $\omega$.

Under an injectivity assumption for any input space region $\omega$ and $S = \{\boldsymbol{A}_\omega \boldsymbol{z} + \boldsymbol{b}_\omega \forall \boldsymbol{z} \in \omega\}$, according to Theorem 1. in Humayun et al. (2022a), the output density on $S$, $p_S(\boldsymbol{x}) \propto \frac{1}{e^{\psi_\omega}}$. Therefore, local scaling $\psi_\omega$ is proportional to the negative log-likelihood of the generative model for any $\boldsymbol{z} \in \omega$. We can validate this by using a kernel density estimator (KDE) to estimate the density of generated samples on the data manifold from a uniform latent distribution Fig. 2 In fig. 3 for a toy dinosaur manifold and fig. 32 for a toy mixture of gaussians, we demonstrate how the local scaling is lower around the modes of the learned distribution and higher for anti-modes.

### 2.1.2 LOCAL RANK, $\nu$.

The second descriptor we study is the rank of the region-wise slope matrix $\boldsymbol{A}_\omega$, which represents the dimensionality of the manifold learned by a CPWL generator.

**Definition 2.** For a CPWL manifold produced by generator $\mathcal{G}$, *local rank $\nu_\omega$* is the exponent of the Shannon entropy of the spectral distribution of the per-region affine slope $\boldsymbol{A}_\omega$ and can be expressed as:

$$\nu_\omega = \exp\left(-\sum_i^k \alpha_i \log(\alpha_i)\right) \tag{3}$$

$$\text{where } \alpha_i = \frac{\sigma_i}{\sum_i^k \sigma_i} + \epsilon. \tag{4}$$

Here, $\{\sigma_i\}_{i=0}^{i=k}$ are non-zero singular values of $\boldsymbol{A}_\omega$ and $\epsilon = 10^{-30}$ is a constant. The local rank $\nu_\omega$ can be shown to be equivalent to the dimensionality of the tangent space on the data manifold at $\boldsymbol{z}$.

### 2.1.3 LOCAL COMPLEXITY, $\delta$

An important geometric notion to characterize any manifold locally is the local smoothness of the manifold. However, smoothness requires computing the hessian of the input-output mapping making it computationally intractable for large generative models. We therefore consider *local complexity* as a proxy for *sharpness* of the manifold locally for our study. Based on the notion of complexity for CPWL neural networks (Hanin & Rolnick, 2019), we can define local complexity of a CPWL generator as the following.

**Definition 3.** For a CPWL generator with input partition $\Omega$, the *local complexity $\delta_{\boldsymbol{z}}$* for a $P$-dimensional neighborhood of radius $r$ around latent vector $\boldsymbol{z}$ is

$$\delta_{\boldsymbol{z}} = \sum_{\forall \omega \cap V_{\boldsymbol{z}} \neq \emptyset} \mathbb{1}_\omega \tag{5}$$

$$\text{where } V_{\boldsymbol{z}} = \{\boldsymbol{x} \in \mathbb{R}^E : ||\mathbf{B}(\boldsymbol{x} - \boldsymbol{z})||_1 < r\}. \tag{6}$$

Here, $\mathbf{B}$ is an orthonormal matrix of size $P \times E$ with $P \leq E$, $||.||_1$ is the $\ell_1$ norm operator and $r$ is a radius parameter denoting the size of the locality to compute $\delta$ for. Here we consider a $P$ dimensional neighborhood instead of the full dimensionality of the latent space to reduce computational complexity. The sum over regions $\omega \in V_{\boldsymbol{z}}$ requires computing $\Omega \cap V_{\boldsymbol{z}}$ which can be computationally intractable for high dimensions. A proxy for computing the partition for $V_{\boldsymbol{z}}$ with small $r$ is counting the number of non-linearities within $V_{\boldsymbol{z}}$, since for small $r$, the one can assume that the non-linearities do not fold inside $V_{\boldsymbol{z}}$, therefore providing an upper bound on the number of regions according to Zaslavsky's Theorem (Zaslavsky, 1975). To compute local complexity, we use the method introduced by Humayun et al. (2024) for general neural networks. In appendix C we discuss the relationships between the geometric descriptors and present examples for a VAE trained on MNIST.

## 2.2 Extending beyond Continuous Piecewise-Linear Generators

**Computing jacobians for large networks.** For any latent vector $\boldsymbol{z} \in \omega$, $\boldsymbol{A}_\omega$ can be obtained by computing the input-output jacobian of the network. Computing the singular values of the full input-output jacobian is significantly expensive for large networks. Therefore, when computing local scaling and rank we obtain singular values via randomized SVD Halko et al. (2011). First we obtain a random projection matrix with orthonormal rows $\mathbf{W}$ with shape $k \times n$ such that $\mathbf{WW}^T = \mathbb{I}_k$. Here $n$ is the dimensionality of the outputs generated by the network. We therefore approximate local scaling as:

$\psi_\omega^{(trunc)} = \sum_{i=1}^{k} log(\sigma_i^{(trunc)})$, where $\sigma_i^{(trunc)}$ are the non-zero singular values of $\mathbf{WA}_\omega$.

For any $\omega$, if $\mathbf{W}$ forms a basis for the range of $\mathbf{A}_\omega$ then $\sigma_i \approx \sigma_i^{(trunc)} \forall i = 1, 2 \ldots k$ [4]. Therefore $\mathbf{WA}_\omega$ would provide us a low-rank approximation of $\mathbf{A}_\omega$.

In our experiments we have tried two methods to obtain the projection matrix $W$ 1) by obtaining the eigenvectors for the covariance matrix for a set of 50K randomly generated samples. This was suggested in Halko et al. (2011). 2) by performing QR decomposition of a randomly initialized matrix. We see that the performance difference between methods 1) and 2) are negligible therefore consider the cheaper alternative 2) and consider a fixed pre-computed $\mathbf{W}$ with k=120 for all $\mathbf{A}_\omega$ in our Stable Diffusion experiments.

**Networks with smooth activations.** While the descriptors are defined for CPWL mappings, modern generative models employ a mixture of CPWL and non-CPWL operations. For networks with smooth activation functions or non-piecewise-linear non-linearities, our descriptors are first order Taylor approximations. For example, Stable Diffusion employs the GeLU activation function for which we perform the bulk of our experiments in following sections. Smooth activation functions induce a soft VQ partitioning of the latent space compared to the hard VQ partitioning induced by a CPWL map Balestriero & Baraniuk (2018b), retaining much of the local linear structure we expect in CPWL maps. Recent work has also empirically verified the local linearity for a large class of image based diffusion models Chen et al. (2024) suggesting the reliability of first-order approximations.

## 3 Characterizing the Local Geometry of Pre-Trained Models via Descriptors

In this section, we explore the geometry of pre-trained generative models by characterizing the latent space to output manifold mapping in terms of the local geometric descriptors mentioned in the previous section. We are interested in the following questions: i) How does the on manifold local geometry vary from the off manifold local geometry? ii) How does the local geometry vary across the input domain?

### 3.1 On and Off Manifold Geometry for Denoising Diffusion Probabilistic Models

**Setup.** To study the on and off manifold geometry of diffusion models, we train a denoising diffusion probabilistic model (DDPM) (Ho et al., 2020) on a toy dataset[1] to visualize how the local geometry varies for 1) different noise levels $t$, and 2) different training iterations.

---

[1] https://jumpingrivers.github.io/datasauRus

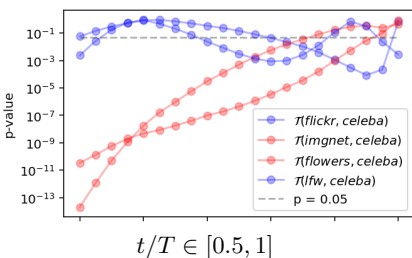

Figure 3: Local geometric descriptors computed over the input domain of a DDPM trained on samples from a toy dinosaur manifold $\mathcal{M} \in \mathbb{R}^2$, conditioned on $t = 0.22T$ (left three columns). Denoising dynamics of local descriptors (right three columns) for different number of training optimization steps. *For a well trained diffusion model, local descriptors can distinguish between on and off manifold vectors in the input space.* See fig. 15 for more.

Figure 4: **Statistical significance of the difference between local scaling distributions** for in-domain (blue) vs out-of-domain (red) datasets when conditioned on different noise levels for an SD Unet trained on the CelebAHQ dataset. Here $\mathcal{T}(a, b)$ denotes a t-test between local scaling distributions for dataset $a$ and dataset $b$.

In Fig. 3-heatmaps we present the local complexity $\delta_x^t$, local scaling $\psi_x^t$ and local rank $\nu_x^t$ computed for different input space vectors $x$ using the DDPM conditioned on noise levels $t$. Here $T$ is the highest noise level in the forward diffusion process. We also present the difference between the expected descriptor values on and off the manifold, $\mathbb{E}_{\mathcal{M}}[\Phi] - \mathbb{E}_{\bar{\mathcal{M}}}[\Phi], \forall \Phi \in \{\psi^t, \delta^t, \nu^t\}$ at different training iterations (right). We consider the set of input vectors within $0.05$ units of the training data as on manifold $\mathcal{M}$ and rest as off the manifold $\bar{\mathcal{M}}$.

**Observations.** The first observation is that with longer training, the maximum absolute difference between on and off manifold local geometry $\max_t\{|\mathbb{E}_{\mathcal{M}}[\Phi] - \mathbb{E}_{\bar{\mathcal{M}}}[\Phi]|\}$ increases. *Since with more training we see higher distinction between the on and off manifold geometry, this difference can be an indicator of learning in diffusion models.* We see that for well trained models, apart from $t > 0.17T$, $\psi_x^t$ and $\nu_x^t$ decreases and $\delta_x^t$ increases with decreasing $t, \forall x \in \mathcal{M}$.

This means, the likelihood on the manifold increases as noise levels are reduced, the smoothness decreases and the dimensionality of the manifold decreases as well. The quantity $\mathbb{E}_{\mathcal{M}}[\Phi] - \mathbb{E}_{\bar{\mathcal{M}}}[\Phi]$ is also minimized at $t \approx 0.17T$. This indicates that there can exist a noise level $t$ conditioned on which diffusion model local scaling, rank and complexity have the highest distinction geometrically between on and off manifold vectors from the input space. In fig. 4, we see statistically significant difference between the on vs off manifold geometry at smaller $t$ for Stable Diffusion (SD). *These result indicate that the local geometry can allow directly probing which parts of the input space are on the learned manifold* to possibly perform one step denoising or propose novel guidance schedules.

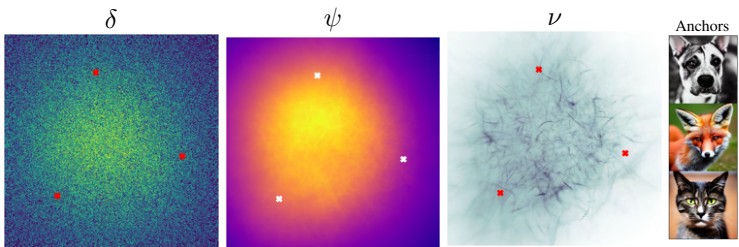

Figure 5: **Geometry of the Stable Diffusion latent space.** Geometric descriptors (left, middle-left, middle-right) visualized on a 2D latent space subspace, that passes through the latent representations of "a fox", "a cat" and "a dog" (right), denoted via markers on the 2D subspace descriptor. In Appendix, we provide denoised images for different high/low descriptor regions from the subspace. We see that in the convex hull of the three anchor latent vectors $\psi \uparrow$, $\nu \downarrow$ and $\delta \uparrow$. Moreover we see that in the convex hull, the local rank $\nu$ undergoes sharp changes which are not visible towards the edges of the domain.

## 3.2 THE LOCAL GEOMETRY OF LATENT DIFFUSION MODELS

In Sec. 3.1, we see that the local geometry in the input domain of a ddpm can be distinctive of its learned manifold. In this section we study the local geometry of the Stable Diffusion (SD) latent space, to explore whether there exists a relationship between the local geometry and the domain of the SD decoder.

**Setup.** While in Sec. 3.1 we could visualize the whole input domain of the diffusion model, for SD we can only visualize a subspace of the SD latent space. We use three prompts "a cat", "a dog" and "a fox" to generate three latent vectors using the SD diffusion model and consider a 2D slice in the latent space, going through the three denoised latents as our domain to visualize. Note that since this is a 2D subspace of the latent space, we can expect part of it to be in-domain for the SD decoder, whereas part of it would be out-of-domain.

**Observations.** We observe that 1) In the convex hull of the three denoised latents used as anchors for the 2D subspace being visualized, we have higher complexity, lower rank and higher local scaling. The decoded images from the convex hull may contain artifacts but are legible generations. 2) Local rank does not smoothly vary across the latent space, especially with sharp changes in the local rank within the convex hull of the anchor latent vectors. For the lowest rank regions in the convex hull, decoded images have good fidelity compared to latents with high uncertainty or complexity. 3) If we move away from the convex hull, we see that generated images become more broken and contain heavy artifacts, indicating that such regions are out-of-domain for the SD decoder. However, we see that the local scaling is lower in these regions compared to the convex hull.

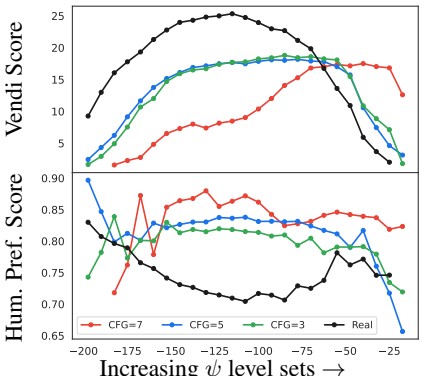

Figure 6: **Local Geometry level sets Imagenet prompts.** Vendi diversity scores and RAHF Liang et al. (2024) aesthetic scores computed for images with classifier free guidance (CFG) 7, 5 and 3. Diversity per level set increases and then decrease with increased local scaling. Aesthetic score slightly increases and then decreases as well with increased local scaling.

## 4 SECRETS THAT YOUR MANIFOLDS HOLD

**Visually complex images have higher local scaling**

We selected $20K$ samples from Imagenet with resolution higher or equal to $512 \times 512$, encode the samples using the SD encoder, and compute the local descriptors for the SD decoder. In Appendix Fig. 17 each column represents a local scaling level set, with the $\psi$ for columns increasing from left to right. Recall that local scaling is proportional to the negative log-likelihood. In Appendix Fig. 17, we can see that for lower local scaling images we have more modal features in the images, i.e., the samples have less background elements and are focused on the subject corresponding to the Imagenet class.

For images with higher local complexity, we see more qualitatively outlier characteristics and higher visual complexity. For images with higher local rank in Appendix Fig. 19, we see that the backgrounds have higher frequency elements compared to lower rank images. For higher rank images,

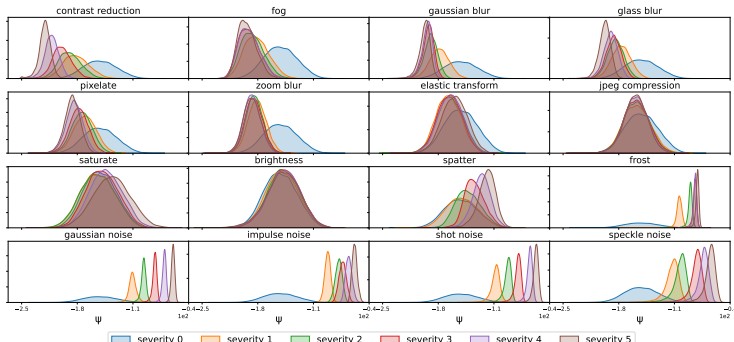

Figure 7: **Local scaling is sensitive to image corruptions.** We use 16 corruptions from (Hendrycks & Dietterich, 2019) to corrupt $10K$ imagenet images and compute the local geometry of the SD decoder. We see that SD local geometry is sensitive to corruptions, i.e., aesthetic changes to in-domain images (here Imagenet).

the dimensionality of the manifold is higher locally, therefore allowing more noise dimensions on the manifold. *These results indicate that the local geometry is indeed sensitive to qualitative variations such as visual complexity of Imagenet images.*

**Noise corruptions increase and blurring corruptions decrease local scaling**

Fig. 7 illustrates the effect of applying 16 different image distortions (originally proposed in (Hendrycks & Dietterich, 2019)) to 10k ImageNet images. We consider Imagenet as in-domain for Stable Diffusion and encode them to the SD latent space to compute the geometric descriptors for the SD decoder. Samples are uniformly distributed over its classes. The plot shows the local scaling distribution at 6 increasing levels of severity $\in \{0, 1, 2, 3, 4, 5\}$, with zero corresponding to no corruptions applied. We observe that corruptions that are associated with reduction of spectral band, and/or reduction to the color range result in a reduction to the local scaling therefore the negative log-likelihood. We conjecture that this is due to the averaging effect of such distortions which move the corrupted images close to the mean of all images. Conversely, distortions known to be associated with the introduction of high-frequency artifacts are observed to produce an increase in local scaling therefore uncertainty moving the images away from the mean. *The results clearly indicate that the local geometry is sensitive to aesthetic changes to images introduced via most of the 16 corruptions.*

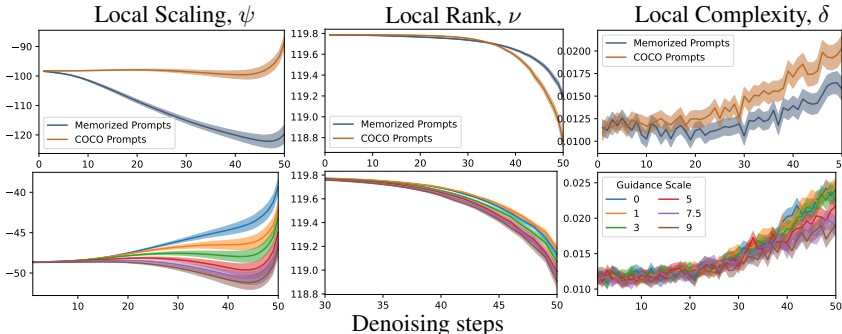

Figure 8: **Local geometry of denoising trajectories.** Geometric descriptors computed for the SD decoder unconditionally, during 50 stable diffusion denoising steps, for (top) 100 COCO and 100 memorized prompts (Wen et al., 2024) with guidance scale 7.5 and (bottom) 100 COCO prompts with varying guidance scales. For each prompt or guidance scale, we start from the same seeds. Shaded region represents 95% confidence interval. *We see that the local geometry trajectories are discriminative of memorization, as well as increased alignment when stronger classifier free guidance is used.*

**Memorized and aligned denoising trajectories are locally contracted and smooth.**

Classifier-free-guidance is a method for increasing the alignment between the conditioning text-prompt and generated image Rombach et al. (2021). In Fig. 8-bottom, we see that for higher classifier free guidance scales during denoising, the avg. local scaling is lower, avg. local rank is lower and the avg. local complexity is higher. This indicates that for more conditionally aligned images obtained via classifier-free-guidance of the SD diffusion Unet, the decoder uncertainty is also lower especially during the final denoising steps. We also compute the local geometric descriptors for denoising trajectories conditioned on memorized prompts Wen et al. (2024) vs coco captions (Fig.8-top). We see that the mean local geometry is significantly different for denoising trajectories of 100 memorized prompts vs 100 random coco prompts. We also see that for higher guidance scales local rank $\nu$ is $\downarrow$ and local complexity $\delta$ is $\downarrow$ as well.

**Connections with generation diversity and human preference scores.**

In fig. 6 we present Vendi score Friedman & Dieng (2023) and human preference score Liang et al. (2024) (higher is better) aggregates for $50K$ real and generated Imagenet images with classifier free guidance scales of 7, 5 and 3. The images are sorted in increasing local scaling $\psi$ bins from left to right. We see that for local scaling level sets from the lowest to the middle bins, we have an increase in the diversity of images per bin. For the highest local scaling bins, we get images from the highest uncertainty modes, i.e., the anti-modes, which result in a drop in the diversity of images (see Appendix fig. 32 for a toy illustration). We observe similar trends in diversity for an Imagenet

trained Diffusion Transformer (DiT) Peebles & Xie (2023) discussed in Appendix appendix D.1. For generated images we see human preference score marginally increasing from left to right. For the highest local scaling bins, especially for lower CFG, we see a drop in the human preference scores. For real images, human preference scores have a significantly different trend compared to generated images. Note that the RAHF human preference model Liang et al. (2024) is trained on synthetic images and therefore might be less reliable for real images.

## 5 GUIDING GENERATION USING GEOMETRY AS A REWARD

In the previous sections, we have presented qualitative and quantitative evidence, establishing the connection between geometric descriptors and downstream generation. Among the three descriptors we find that local complexity has the highest sensitivity to aesthetic changes in images due to corruptions (fig. 7), and correlates with visual complexity (fig. 17). We also observe in fig. 6 higher local scaling level sets for samples generated using a classifier free guidance scale of 7.5, have higher diversity while maintaining higher predicted human preference scores. Based on these results, we wish to explore whether local scaling can be used to guide generation.

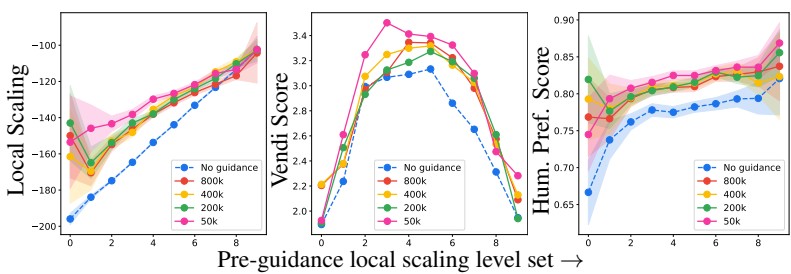

Figure 9: **Guidance via reward models trained with increasing number of training samples.** We observe that even with when only 50K samples are used in training, reward models can increase generation diversity, local scaling and human preference scores. Here we present the change of vendi score, average local scaling and average aesthetic score for pre-guidance local scaling level sets increasing from left to right on the x-axis.

Recently proposed instance-level universal guidance method (Bansal et al., 2023), can effectively influence the latents in the reverse process of a latent diffusion models to produce desired changes. Directly using local scaling to guide generation using such methods, require calculating the input-output Hessian since local scaling is a first-order measure that requires comput-ing the input-output jacobian. To avoid computing the Hessian we train a reward model as a proxy and use the reward model gradients directly. Instead of training on continuous local scaling values in a regression task, we transform it into a local scaling level set classification task. We discretize the range of local scaling values into 5 bins and use the bin indices as training labels.

**Data preparation.** We obtain training data for the reward model by i) sampling $N$ images from Imagenet and encoding them to the Stable Diffusion latent space ii) adding noise using the forward diffusion process up to randomly chosen noise levels iii) for each latent computing the local scaling descriptor.

To evaluate the performance of the reward model and dependency of the reward model on the number of training samples, we train multiple models for $N = 50K, 200K, 400K, 800K$ . For evaluation, we generate 2560 samples using the dreambooth live subject prompt templates Ruiz et al. (2023), with Imagewoof Howard (2019) dogs as subjects. While Imagewoof dog classes are present in Imagenet therefore possibly in the training data, the dreambooth prompt templates contain a variety of settings that are not generally present in Imagenet, e.g., 'a <subject> on top of pink fabric'.

**Evaluation Setup.** We first sample Stable diffusion without any reward guidance and with classifier-free guidance of 7.5 to obtain baseline samples. We partition the range of local scaling values obtained for the baseline samples into $n = 10$ bins fig. 9, where each bin contains images from a local scaling level set. Following that we use the same seed and prompts as the baseline samples to generate images using reward guidance to increase local scaling. For each bin or pre-guidance local scaling level set, we compare between the baseline samples and corresponding reward guided generations in the following three axes: i) change of local scaling ii) change of vendi (diversity) score iii) the change of human preference score (RAHF Liang et al. (2024) aesthetic score).

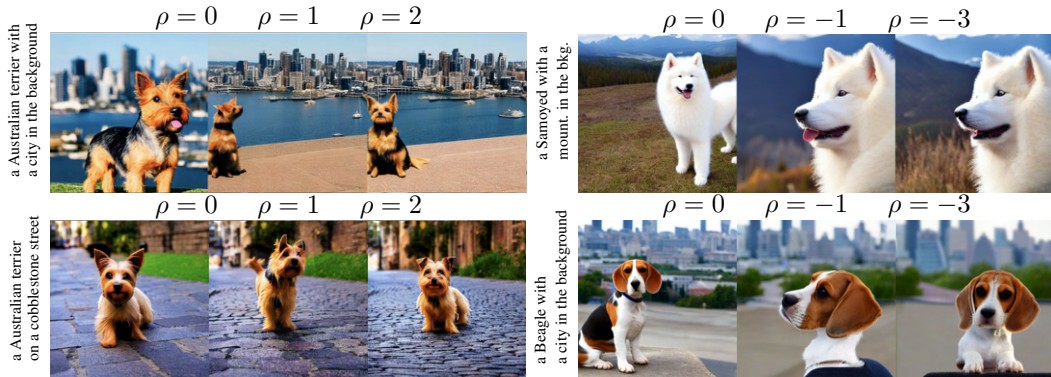

Figure 10: **Reward guidance on stable diffusion.** Local scaling reward guidance increases from left to right in each picture (left-panel) and decreases left to right in each picture (right-panel), with the first image showing no reward guidance. We observe maximizing the reward leads to sharper details, improved sharpness and contrast, and higher diversity in the images. Decreasing the local scaling leads to minimized uncertainty, resulting in a noticeable blurring effect and loss of details especially in the background of the image.

**Results.** In fig. 9, we present the mean local scaling per bin with 95% confidence interval. Here the blue line represents the mean pre-guidance local scaling values, increasing from left to right. In fig. 9, we present vendi scores and average predicted human preference scores. For any bin, we present results for the reward guidance scale that maximizes the local scaling. We see that even for a model trained with $50K$ samples, we can have a considerable increase in the local scaling, diversity and aesthetic score for most of the bins. Changes in local scaling, vendi and aesthetic scores are higher for the lower pre-guidance local scaling level set bins compared to the higher pre-guidance local scaling level set bins.

Our experiments reveal that maximizing local scaling in the manifold of a stable diffusion model directly correlates with adding texture to the generated images. Moreover, this approach reduces the likelihood on the manifold for single images. By optimizing the local scaling descriptor, the generative model is guided towards producing more varied and textured outputs.

This approach is notable because traditional methods for diversity guidance generally function at the distribution level. Our method, however, focuses on maximizing the inherent diversity as preserved by the model within its learned manifold, effectively steering the generated images towards the extremities of the distribution. This instance-level intervention allows for a more detailed and precise enhancement of diversity, presenting a novel approach to guiding generative models.

As seen from Fig. 10 (left-panel) maximizing the reward results in added details in form of sharpening the image, adding texture and contrast. We also observe that if we move towards minimizing the reward, the images tend to loose fine-grained details as seen in Fig. 10 (right-panel). Please refer to the supplementary material for more visual results.

## 6 CONCLUSION & FUTURE DIRECTIONS

In this paper, we present empirical evidence that the local geometric descriptors – local scaling ($\psi$), local rank ($\nu$) and local complexity ($\delta$) - can effectively characterize the local geometry and distinguish between downstream qualitative aspects of generated samples such as generation quality, aesthetics, diversity, and memorization. Such descriptors only utilize the model's architecture and weights to characterize the behavior of generative models. We acknowledge two main limitations that warrant further investigation. First, the geometry of the learned manifold is inherently influenced by the training dynamics of the model. A deeper understanding of this relationship is needed to fully leverage geometric analysis for models. Second, the computational complexity of our method, particularly the calculation of the Jacobian matrix, may pose a practical challenge, especially for large-scale models. Future work should explore more efficient algorithms or approximations to address this limitation.

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

## A  COMPUTATION TIMES FOR LOCAL DESCRIPTORS

The computation times for the local scaling and local rank computation (since both require one randomized SVD computation for one latent vector) ends up being 3929s for 1000 samples. For local complexity we require 113s for 1000 samples. All the estimates are for a JAX implementation of Stable Diffusion on TPUv3.

Note that to train a reward model, we require the descriptors to be computed only once for each pre-trained model. If we compute the local scaling for 100k samples we require 173.1 TPU v3 hours which is equivalent to 54.58 V100 hours (according to Appendix A.3 Dhariwal & Nichol (2021)). Compared to 79,000 A100 hours required for Stable Diffusion training[2], 24000 hours with enterprise level optimization[3], the computation required for the descriptors and reward model training is significantly small. The computation time for the local descriptors can be further reduced by using a smaller $k$ for our projection matrix $W$, or by using non-jacobian based methods, e.g., estimating the local scaling by measuring the change of volume for a unit norm $\ell_1$-ball in the input space. We leave exploration of these directions for future work.

## B  RELATED WORKS

**Local geometry pre-diffusion.**  Early applications of the local geometry of generative models involved improving the generation performance and/or utility of generative models via geometry inspired methods. For example, in Rifai et al. (2011) the authors proposed regularizing the contraction of the local geometry to learn better representations in autoencoders trained on MNIST and CIFAR10. The regularization penalty is employed via the norm of the input-output jacobian in Rifai et al. (2011), is an upper bound for local scaling presented in our paper. In Arvanitidis et al. (2017) the authors provided visualizations on the curvature of pre-trained VAE latent spaces and proposed using an auxiliary variance estimator neural network to regularize the latent space geometry during generation. In Kuhnel et al. (2018) the authors perform latent space statistical inference problems, e.g., maximum likelihood inference, by training a separate neural network to approximate the Riemannian metric. In Humayun et al. (2022a) the authors proposed a novel latent space sampling distribution based on the latent space geometry that allows uniformly sampling the learned data manifold of continuous-piecewise affine generators. The authors showed downstream benefits with fairness and diversity for such latent space samplers. While most of these methods discuss pre-diffusion architectures, their results are early demonstrations of how the local geometry can affect downstream generation. also employ auxiliary Neural Networks to model an intrinsic property of a pre-trained generator, similar to how we propose using a reward model for Stable Diffusion.

**Local intrinsic dimensionality of diffusion models.** The local geometry of diffusion models and possible applications have garnered significant interest in recent years. In Stanczuk et al. (2022) the authors propose a method to compute the intrinsic dimensionality of diffusion models using the assumption that the score field is perpendicular to the data manifold. For any vector $x$ on the data manifold, the method requires computing the dimensionality of the score field around $x$ and subtracting it from the ambient dimension. To do that, the authors perform one step of the forward diffusion process $k$ times for $x$, denoise the $k$ noisy samples using the diffusion model and compute the rank of the data matrix containing denoised samples to obtain the intrinsic dimensionality. Compared to this method, we compute the dimensionality directly via a random estimation of the input-output jacobian SVD. We do not require any assumption on the score function vector field being perpendicular to the data manifold, which may not hold for a diffusion model that is not optimally trained or highly complex training datasets like LAION.

In Kamkari et al. (2024) the authors compute rank using the method proposed in Stanczuk et al. (2022) and show that local intrinsic dimensionality can be used for out-of-distribution (OOD) detection. This is analogous to our analysis in Sec 3 on the local geometry on or off the manifold. We can see that the intuition authors provided in Kamkari et al. (2024) for diffusion models trained on smaller models and datasets e.g., FMNIST, MNIST, translate to larger scale models like Stable Diffusion trained on LAION as we have presented fig. 5, fig. 17 and Sec 4. Especially in fig. 7, we show that creating OOD samples with corruptions on Imagenet data (in-distribution), we can have an increase or decrease

---

[2]https://www.mosaicml.com/blog/training-stable-diffusion-from-scratch-costs-160k
[3]https://www.databricks.com/blog/stable-diffusion-2

in negative-log likelihood (estimated via local scaling), with decrease for blurring corruptions and increase in noising corruptions.

Concurrent work Kamkari et al. (2024) has also shown the relationship between the intrinsic dimensionality (local rank) of Stable Diffusion scale models and the texture/visual complexity of generated images. We believe our analysis is much more holistic with three different geometric properties being measured compared to only local dimensinality. We i) show quantitatively how diversity measured via vendi score is higher for higher local scaling and rank values (fig. 6). We have explored how rank and scaling evolves continuously across the latent space in fig. 5. We have presented how the geometry distribution varies as we continually perturb images via noise or blurring operations fig. 7 And finally in Sec 5 we have presented a method to guide generation using the local geometry to obtain downstream generation benefits.

**Misc.** Apart from the aforementioned works, Kadkhodaie et al. (2023) show that the emergence of generalization in diffusion models – when two networks separately trained on the same data learn the same mapping – can be attributed to the eigenspectrum and eigenvectors of the input-output jacobian. While we do not study the training dynamics of the local geometric descriptors in our paper, Kadkhodaie et al. (2023) suggests that the local geometry can be an important indicator of memorization and generalization emergence in diffusion models. In Manor & Michaeli (2023) the authors use the posterior principal components of a denoiser for uncertainty quantification. This work suggests that components with larger eigenvalues result in larger uncertainty which is directly related to the local scaling descriptors as it measures the product of non-zero singular values. While in Manor & Michaeli (2023) the authors propose using it for only a single image denoiser, we show that it generalizes for any diffusion model including Stable Diffusion scale text-to-image diffusion models.

## C    CORRELATIONS BETWEEN THE THREE DESCRIPTORS

*Local scaling* characterizes the change of volume by the affine slope $\mathbf{A}_\omega$ going from the latent space to the data manifold. *Local rank* characterizes the number of dimensions retained on the manifold after the network locally scales the latent space. Both local rank and scaling quantify first order properties of the CPWL operator. *Local complexity* approximates the 'number of unique affine maps' within a given neighborhood Humayun et al. (2024) by computing the number of CPWL knots intersecting an $\ell_1$ ball in the input/latent space. Therefore local complexity is a measure of 'un-smoothness' and quantifies local second-order properties of a CPWL operator.

**Correlations between local scaling $\psi$ and local rank $\nu$.** By definition, local scaling and local rank are correlated, since both characterize the change of volume by the network input-output map at any input space linear region – also evident in eq. (2) and eq. (3). Local scaling is also upper bounded by local rank, $\psi_\omega \leq \sigma_0^{\nu_\omega}$ where $\sigma_0$ is the largest singular value of $\boldsymbol{A}_\omega$. The correlation is evident for our low dimensional DDPM setting presented in fig. 3, local rank and local scaling are highly correlated in their spatial distribution. There are indications suggesting that the correlations persists throughout training as can be seen in fig. 3 rightmost column top and bottom. However in fig. 5, we can see that in the high-dimensional Stable Diffusion latent space, local scaling and rank are correlated but local rank has sharper changes spatially compared to local scaling.

**Correlations between local complexity $\delta$ and rank $\nu$.** There also exist correlations between local complexity and local rank due to the continuity of CPWL maps – between two neighboring linear regions $\omega_1$ and $\omega_2$, the corresponding slope matrices $\mathbf{A}_{\omega_1}$ and $\mathbf{A}_{\omega_2}$ differ by at most one row. Therefore between two neighboring regions $\omega_1$ and $\omega_2$, $|\nu_{\omega_1} - \nu_{\omega_2}| <= 1$. Informally, the local rank in a neighborhood $V$ is lower bounded by the number of non-linearities in neighborhood $V$. This is evident in the empirical results presented in fig. 3 and fig. 5. In both figures, for input space neighborhoods with higher local complexity, we see a decrease in local rank. However, we do not observe sharp changes in local complexity as we observe in local rank in fig. 5. In fig. 3 we see that local rank is more discriminative of the data manifold compared to local complexity. Their training and denoising dynamics differ significantly as seen in fig. 3 rightmost column.

**Qualitative and quantitative results on correlations.** We train a beta-VAE unconditionally on MNIST and present in Fig. 11 samples from increasing local descriptor level sets from left to right along the columns. In Fig. 12, we present joint distributions of local scaling, complexity, rank and mean squared reconstruction error for training and test samples. We see that while local scaling,

complexity and rank have some linear correlation, the classwise distribution in fig. 12 is very different between the three. We also present in fig. 13 the vendi score for increasing local scaling level sets and evidence that the population means for the descriptors don't follow the same pattern between sub-populations.

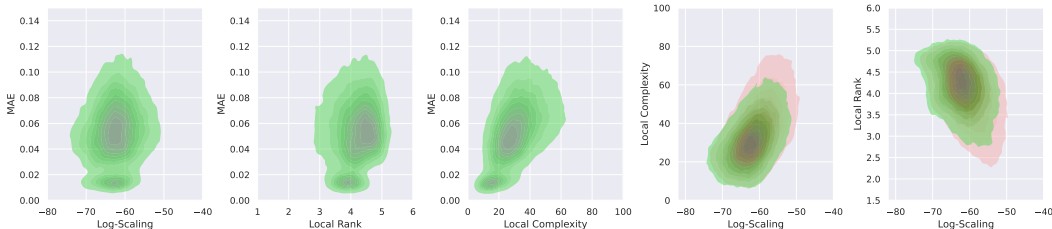

Figure 11: Joint distributions for local scaling and MSE, local rank and MSE, local complexity and MSE, local scaling and local complexity, and local scaling and local rank. We observe that local complexity is linearly correlated wth MSE, with higher complexity images incurring higher error. Local scaling, rank and complexity have correlations between them as well.

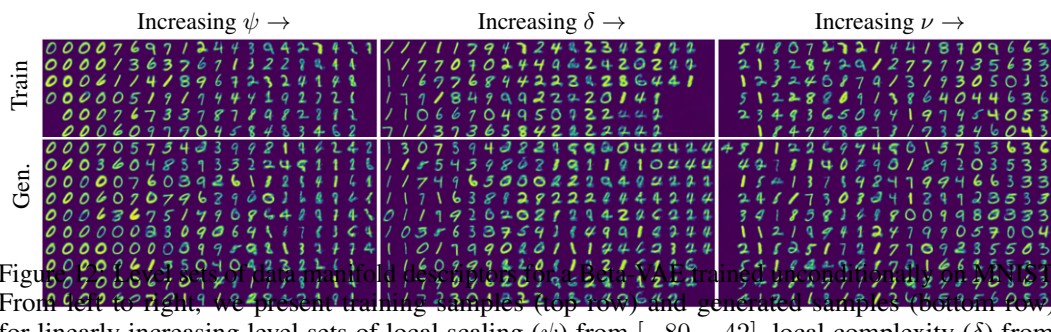

Figure 12: Sorting by $\psi$, $\delta$ and $\nu$ local manifold descriptors on a Beta-VAE trained unconditionally on MNIST. From left to right, we present training samples (top row) and generated samples (bottom row) for linearly increasing level sets of local scaling ($\psi$) from $[-80, -42]$, local complexity ($\delta$) from $[0, 120]$ and local rank ($\nu$) from $[1.5, 5.5]$. Not all level sets had an equal number of samples from training/generated distributions. We see that for higher $\psi$, we have more outlier samples whereas for lower $\psi$ we have modal samples. For increasing $\delta$ we see that the quality of generated samples decreases and the diversity of samples is reduced as well. For higher $\nu$ digits become more regularly shaped.

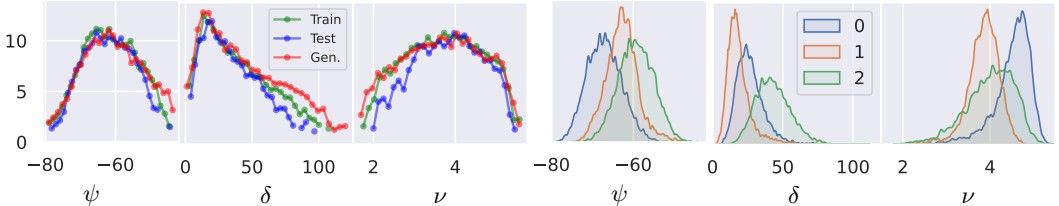

Figure 13: (Left panel) Vendi score (Friedman & Dieng, 2023) calculated for samples from different local descriptor level sets of a Beta-VAE. We take upto $150$ samples from each level set and compute vendi score seperately for the MNIST train dataset, test dataset and generated samples. (Right panel) Sub-population differences of local descriptors in training data. We see that the order of sub-population means for the three classes, are not the same for all three descriptors.

# D  ADDITIONAL EXPERIMENTS

## D.1  LOCAL SCALING FOR TRANSFORMER BASED DIFFUSION MODEL

Since we are based on the CPWL formulation of NNs, our framework would generalize to models of any scale and any architecture with CPWL non-linearities. Empirically we have shown it to generalize for non-CPWL architectures like Stable Diffusion v1.4 and DDPM that employs non CPWL non-linearities such as attention, GeLU and much more. We have performed additional

experiments with a DiT-XL Peebles & Xie (2023) trained on Imagenet-256. For the DiT we compute the descriptors for the transformer network, conditioned on noise level $t = 0$, i.e., zero noise level. We generate 5120 images conditioned on Imagewoof Howard (2019) classes and present in fig. 14, increasing local scaling level sets from left to right. We see that similar to fig. 17 from the, DiT exhibits a qualitative correlation between visual complexity and local scaling. For additional analysis we repeat the Stable Diffusion experiments on the relation between diversity and local scaling for DiT. We see that similar to Stable Diffusion, for increasing local scaling level sets, the diversity of images increase and then drop for the highest local scaling level sets.

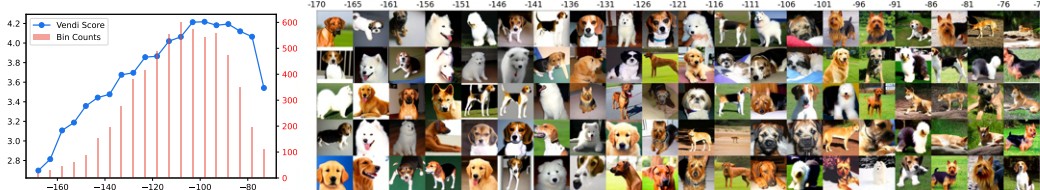

Figure 14: Left: Vendi score and membership counts for increasing local scaling level sets, computed for a DiT transformer. We see that similar to Stable Diffusion, local scaling increases from lower to higher local scaling level sets, then drops for very high local scaling level sets. Right: Generated samples from each level set in the left panel. Sample sets from higher local scaling level sets, tend to be more diverse.

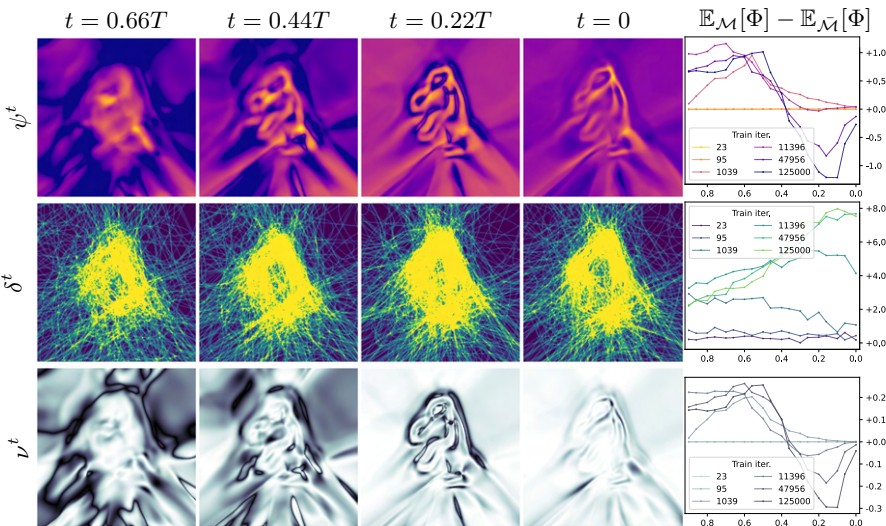

Figure 15: **Local geometric descriptors** computed over the input domain of a pre-trained toy diffusion model trained to produce samples from a dinosaur manifold $\mathcal{M} \in \mathbb{R}^2$. Descriptors are computed by conditioning the diffusion model on noise level $t$. We consider the set of input vectors within $0.05$ units of the training data as on manifold $\mathcal{M}$ and rest as off the manifold $\bar{\mathcal{M}}$. We present the difference between the expected descriptor values on and off the manifold, $\mathbb{E}_{\mathcal{M}}[\Phi] - \mathbb{E}_{\bar{\mathcal{M}}}[\Phi], \forall \Phi \in \{\psi^t, \delta^t, \nu^t\}$ at different training iterations (right). We also present the descriptor computed over $[-6, 6]^2$ for different noise levels $t$ after $125000$ training iterations (rest). We observe that $\psi^t$ is lower, $\delta^t$ is higher and $\nu^t$ is lower on the manifold than off the target manifold for lower noise levels, especially after the model is considerably trained. This indicates that for well trained diffusion model, i.e., learned manifold $\hat{\mathcal{M}} \approx \mathcal{M}$, local descriptors can distinguish between on and off manifold vectors in the input space.

## D.2 VAE TRAINING DYNAMICS FOR MNIST

**Setup.** We train a Variational Auto Encoder (VAE) on the MNIST dataset with width 128 and depth 5 for both encoder and decoder. We add Gaussian noise with standard deviation $\{0, 0.0001, 0.001, 0.01, 0.1\}$ to the training data. Initialization was not kept fixed. In Fig. 16, we

present plots showing the training dynamics of local complexity and scaling, averaged over all test dataset points from MNIST.

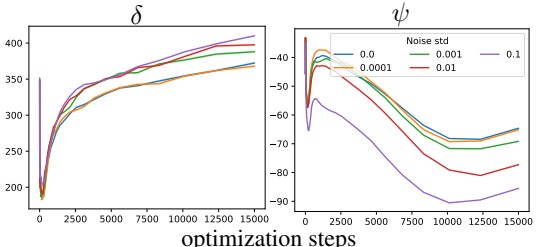

Figure 16: **Training dynamics of geometric descriptors for a VAE trained on MNIST with additive noise.** *As training progresses local complexity $\delta$ increases and local scaling $\psi$ decreases suggesting an increase in expressivity and decrease in uncertainty on the data manifold.* At latter time-steps, $\psi \downarrow$ and $\delta \uparrow$ if noise std. is increased.

**Observations.** By increasing the noise we control the puffiness of the target manifold. We observe that as the noise standard deviation is increased there is 1) increase in $\delta$ indicating the manifold becomes less smooth 2) decrease in local scaling indicating that the uncertainty decreases. We can also observe an initial dip in both local complexity and local scaling. This is similar to what was observed for discriminative models in (Humayun et al., 2024) where a double descent behavior was reported in the local complexity training dynamics of classification models. Based on these results, contrary to the observation in (Humayun et al., 2024), generative models do not have a double descent in local complexity however we do observe a double ascent in local scaling. *Our observations suggest that the training dynamics need to be taken into account, when comparing the local manifold geometry between two separately trained models.*

# E  ENTROPY DIFFERENCE BETWEEN TWO REGIONS

Suppose we have an injective $\mathcal{G} : Z \rightarrow X$ mapping learned by a CPWL generator $\mathcal{G}$. Any linear region $\omega$ in the latent space CPWL partition $\Omega$ is mapped to a unique region on the output manifold. We define $S$ as:

$$S = \mathcal{G}(\mathbf{z}) \forall \mathbf{z} \in \omega = \mathbf{A}_\omega \mathbf{z} + \mathbf{b}_\omega \ \forall \mathbf{z} \in \omega$$

The change of volume from $\omega \rightarrow S$ is $\sqrt{det(\mathbf{A}_\omega{}^T \mathbf{A}_\omega)}$. Therefore for any latent $z$ and output $x = \mathcal{G}(\mathbf{z})$:

$$p_\mathcal{G}(\mathbf{x}) = \sum_{\forall \omega \in \Omega} \frac{p_Z(\mathbf{z})}{\sqrt{det(\mathbf{A}_\omega{}^T \mathbf{A}_\omega)}} \mathbb{1}_{z \in \omega}$$

For any $\mathbf{z_1} \in \omega_1$ the sum from the above equation can be ignored, since for all other regions the value would be zero.

Taking negative log and expectation on both sides the conditional entropy becomes

$$H(p_\mathcal{G}(\mathbf{x_1}); \mathbf{z} \in \omega_1) = H(p_Z(\mathbf{z_1})) + log(\sqrt{det(\mathbf{A}_{\omega_1}{}^T \mathbf{A}_{\omega_1})})$$

For a uniform latent distribution and two regions $\omega_1$ and $\omega_2$, substituting the second term above with $\psi_{\omega_1}$

$$H(p_\mathcal{G}(\mathbf{x_1}); \mathbf{z_1} \in \omega_1) - H(p_\mathcal{G}(\mathbf{x_2}); \mathbf{z_2} \in \omega_2) = \psi_{\omega_1} - \psi_{\omega_2}$$

# F  BROADER IMPACT STATEMENT

Our proposed framework for assessing and guiding generative models through manifold geometry offers several potential benefits to society. By providing a more objective and automated approach, we can significantly reduce the cost and time associated with human evaluation, making the auditing and mitigation of biases in large-scale models more accessible and efficient. This has implications for promoting fairness and equity in AI systems, particularly in domains where biases can have significant societal consequences.

Furthermore, our approach can empower researchers and practitioners to better understand the relationship between the geometry of learned representations and various aspects of model behavior,

such as generation quality, diversity, and bias. This deeper understanding can inform the development of more robust and reliable generative models, leading to advancements in various fields, including art, design, healthcare, and education.

However, we recognize that our approach is not without limitations and potential risks. While it can be a valuable tool for identifying and mitigating biases, it should not and cannot fully replace human annotators, especially in high-risk domains where human judgment and contextual understanding are crucial. Our method focuses on reducing costs and improving the auditing process, but it should not be used as a standalone approach.

Moreover, the increased automation enabled by our approach raises concerns about the potential displacement of human annotators, leading to job losses and economic disruptions. While our method addresses some aspects of model evaluation, it is not comprehensive and cannot assess all facets of model behavior. Therefore, it should be used with caution and in conjunction with other evaluation methods, including human expertise.

## G    EXTRA FIGURES

Increasing $\psi$

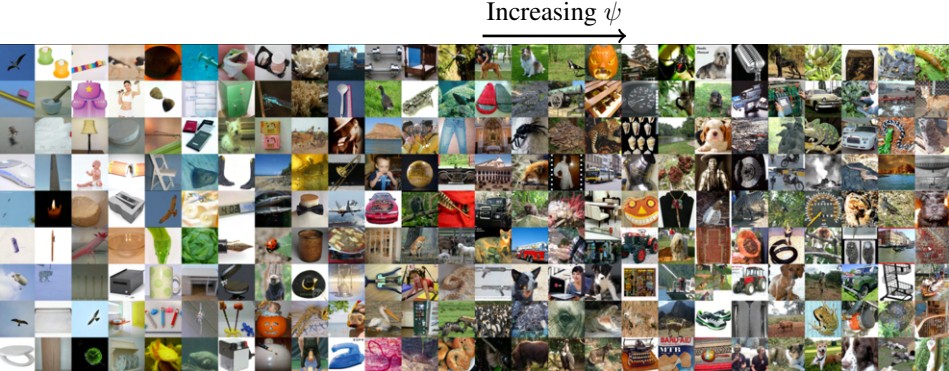

Figure 17: **Local Scaling is sensitive to natural image variations.** ImageNet images ordered along the columns (from left to right), with increasing local scaling $\psi$ of the Stable Diffusion decoder learned manifold. We observe that ImageNet samples with lower values of $\psi$ contain simpler backgrounds with modal representation of the object category. Conversely for higher $\psi$ we have increasing diversity both in background and foreground features.

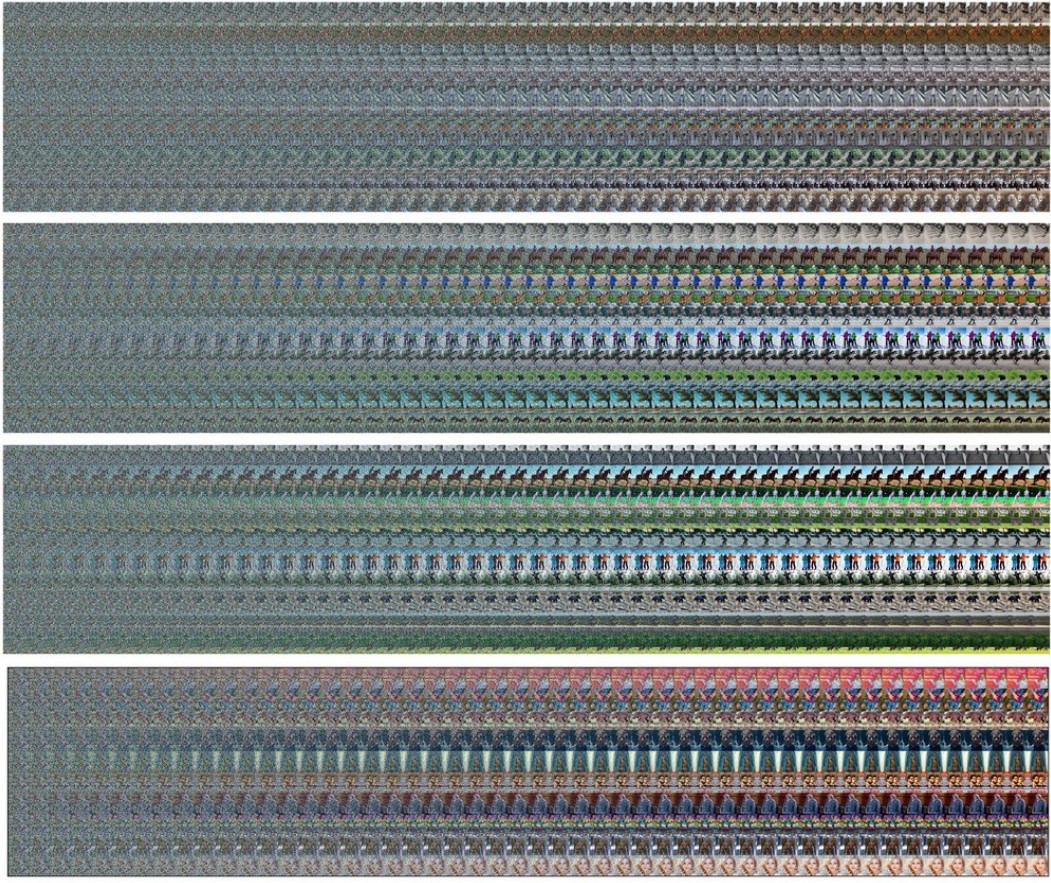

Figure 18: Images generated during 50 diffusion denoising steps for top to bottom, COCO prompts generated with guidance scale 1,5,9 and memorized prompts generated with guidance scale 7.5. Higher guidance scale images, as well as memorized images, tend to resolve faster during the denoising process.

Low local rank, $\nu \downarrow$        High local rank, $\nu \uparrow$

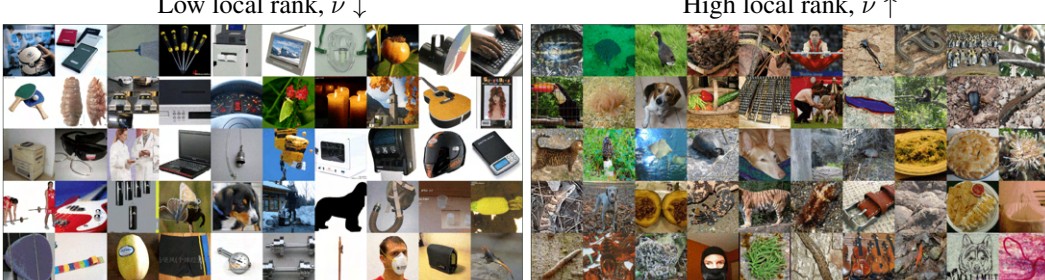

Figure 19: **Influence of the local rank descriptor value on overall image perception.** Images with the lowest (left) and highest (right) local rank $\nu$ from a set of 20000 randomly sampled ImageNet dataset samples. Low rank images contain simpler textures for every class compared to the high rank samples. This is because for images with higher local rank, the learned manifold is higher dimensional therefore allowing higher independent degrees of variations locally for the generated images.

Low local scaling, $\psi \downarrow$        High local scaling, $\psi \uparrow$

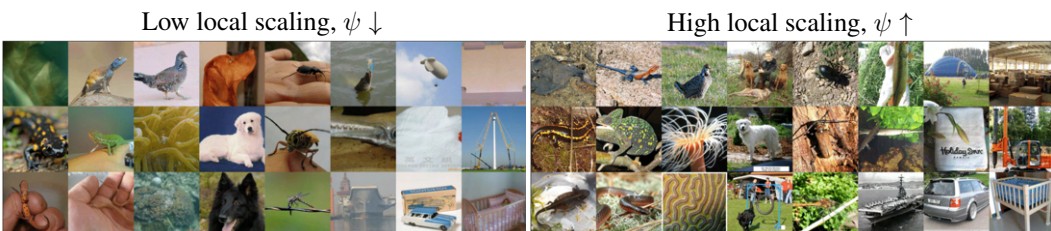

Figure 20: **Influence of the local scaling descriptor.** Imagenet images with high and low local scaling for the stable diffusion decoder. Each coordinate in both left and right image grids, correspond to the same imagenet class.

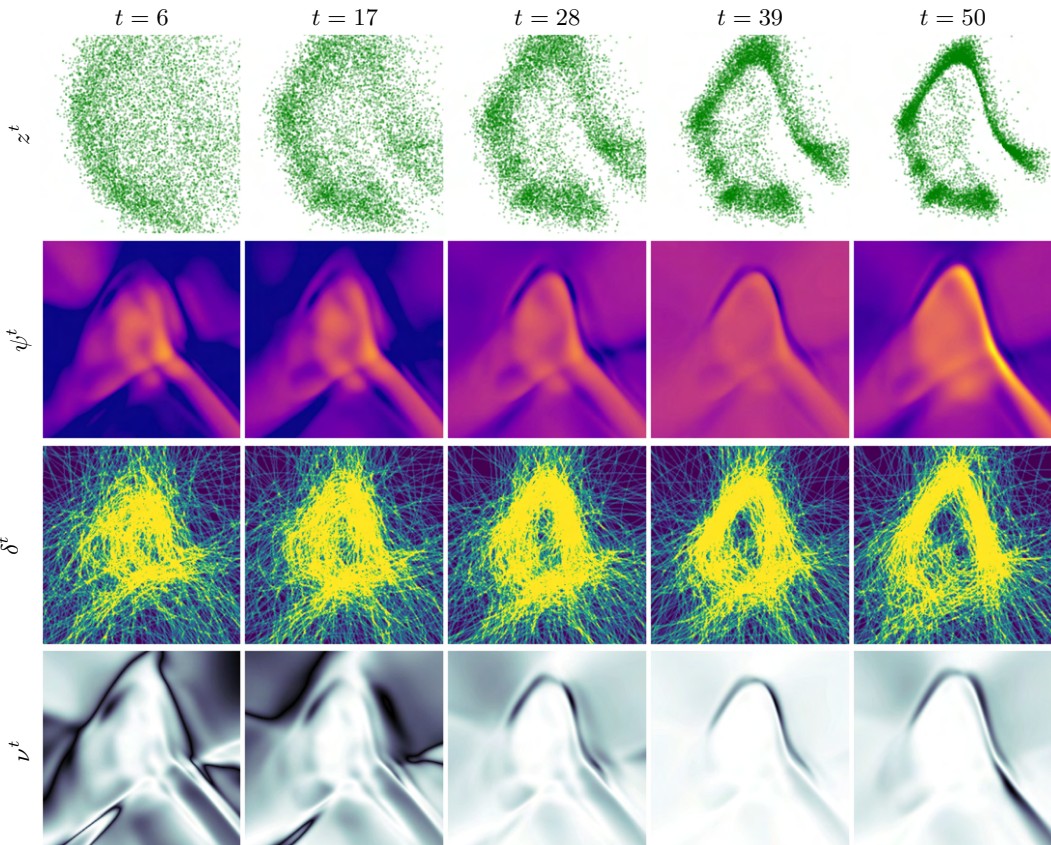

Figure 21: **2D Data Manifold Geometry, A toy Example.** After 11395 optimization steps. Geometry of a diffusion model input-output mapping, trained to on a toy 2D distribution. Local scaling lower around data manifold, local complexity higher around manifold, rank is lower around manifold as well. t=50 has considerably low variance in local scaling showing that final timestep has a diminishing change of density.

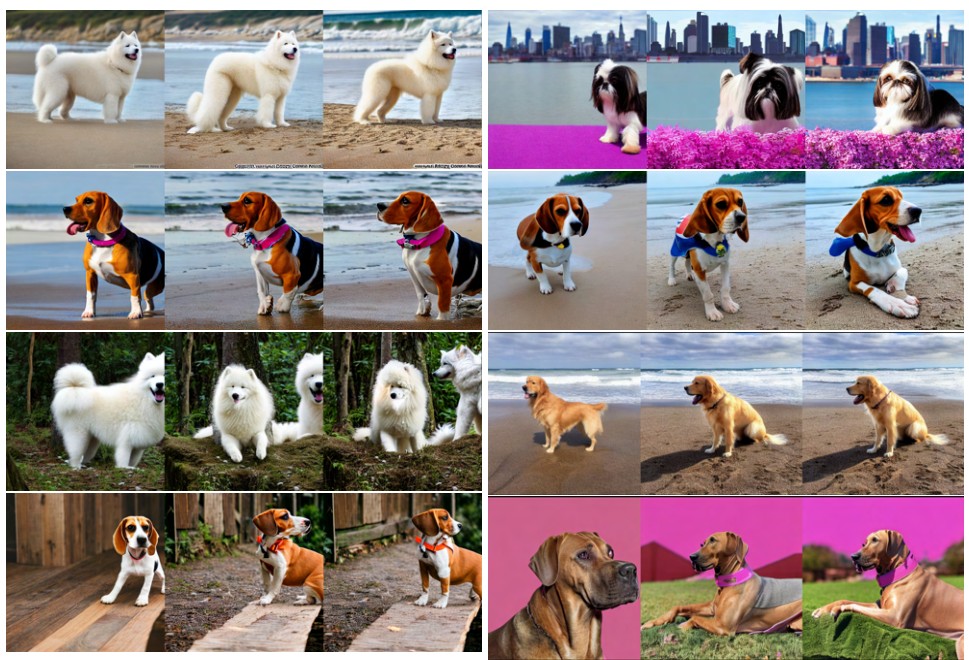

Figure 22: **Reward guidance on stable diffusion (maximizing the reward).**We observe a significant increase in both background detail and artifact diversity within the generated images.

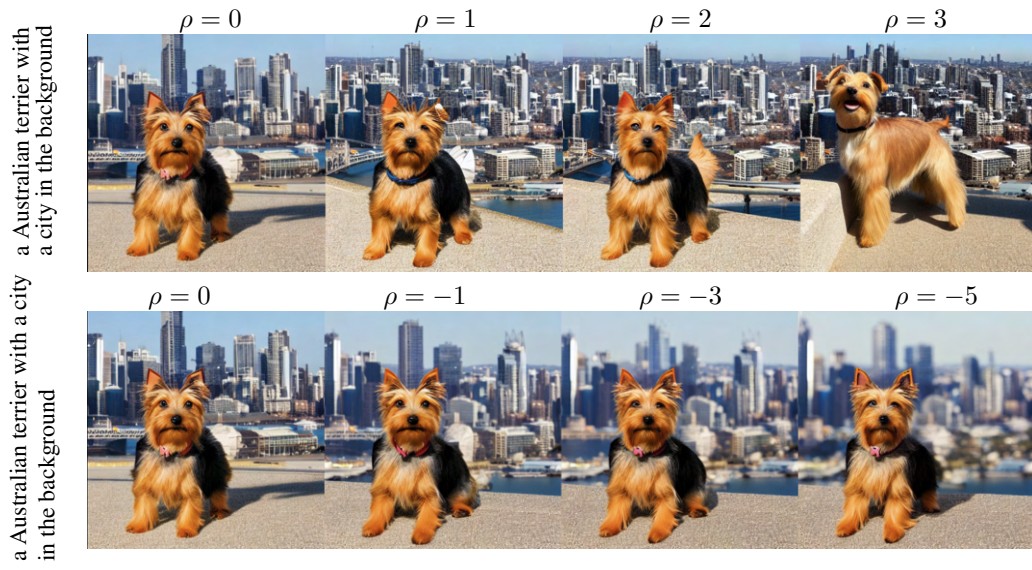

Figure 23: **Controlling mage diversity with local scaling.** Using Reward guidance to increase (top row) and decrease diversity (bottom row) using same initial seed.

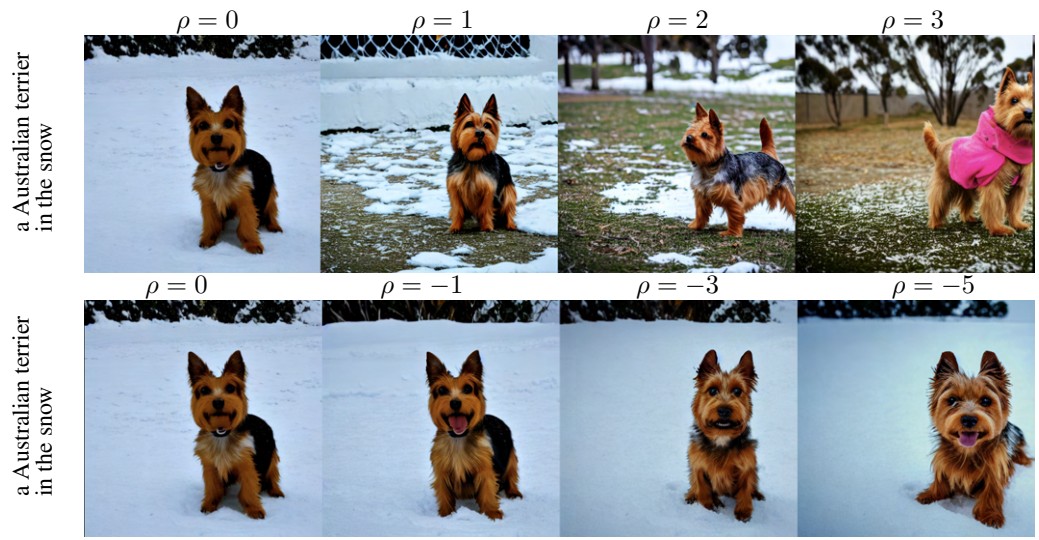

Figure 24: **Controlling mage diversity with local scaling.**Using Reward guidance to increase (top row) and decrease diversity (bottom row) using same initial seed.

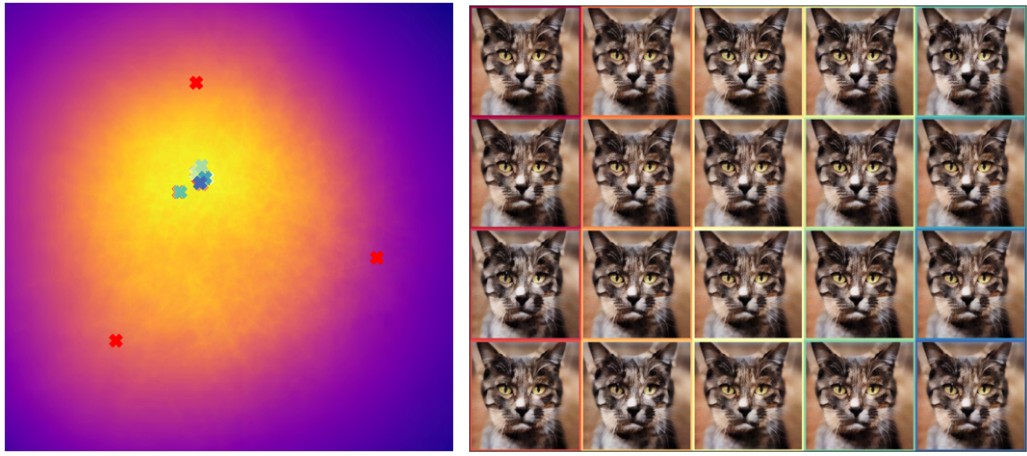

Figure 25: Decoded images (right) using 20 latents (left) from the 2D subspace, with highest $\psi$. Each image bounding box (right) is color coded according to the corresponding latent vector (left).

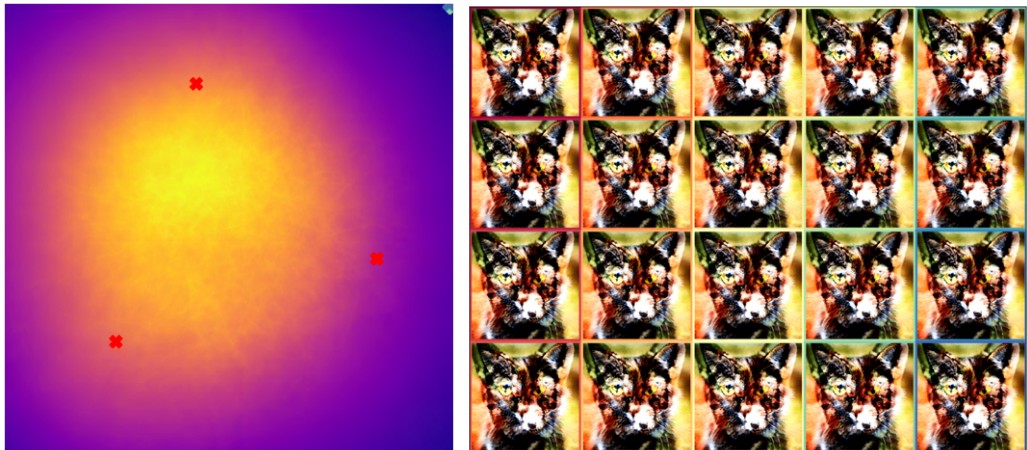

Figure 26: Decoded images (right) using 20 latents (left) from the 2D subspace, with lowest $\psi$. Each image bounding box (right) is color coded according to the corresponding latent vector (left). Selected latents lie outside the domain of the VQGAN latent space.

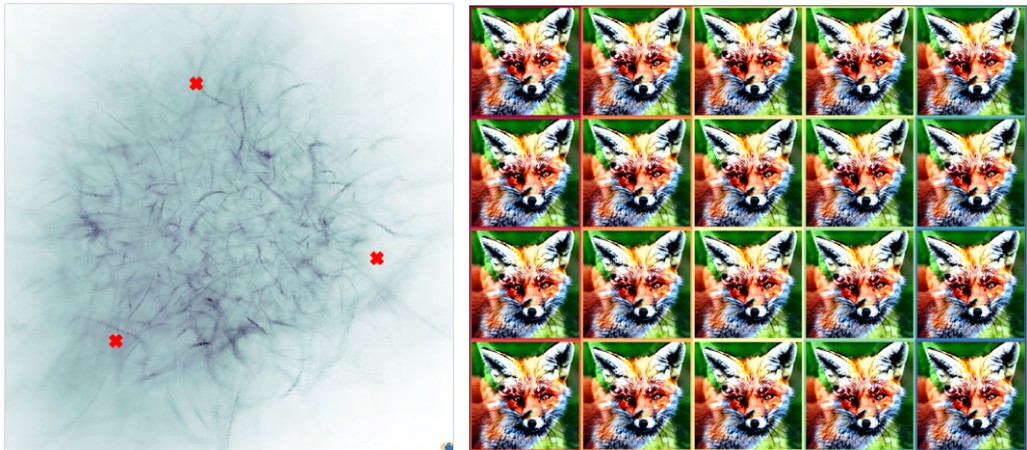

Figure 27: Decoded images (right) using 20 latents (left) from the 2D subspace, with highest $\nu$. Each image bounding box (right) is color coded according to the corresponding latent vector (left). Selected latents lie outside the domain of the VQGAN latent space.

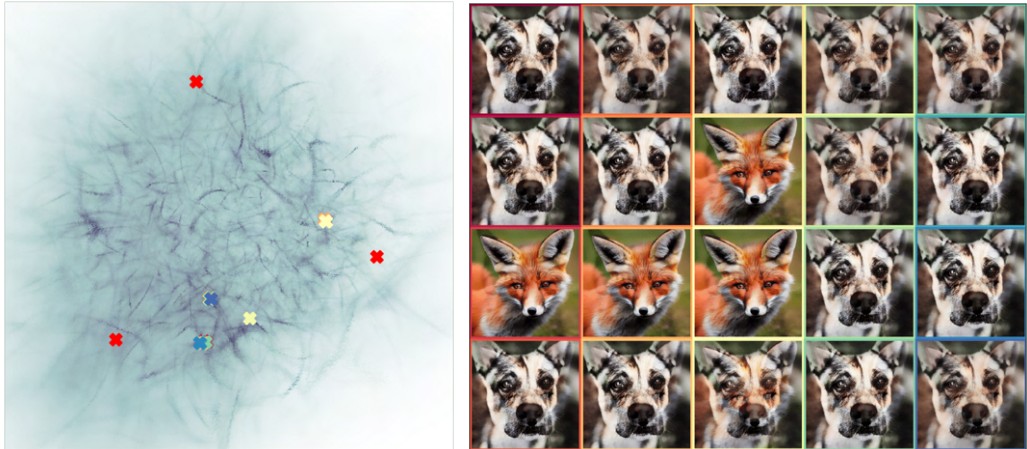

Figure 28: Decoded images (right) using 20 latents (left) from the 2D subspace, with lowest $\nu$. Each image bounding box (right) is color coded according to the corresponding latent vector (left).

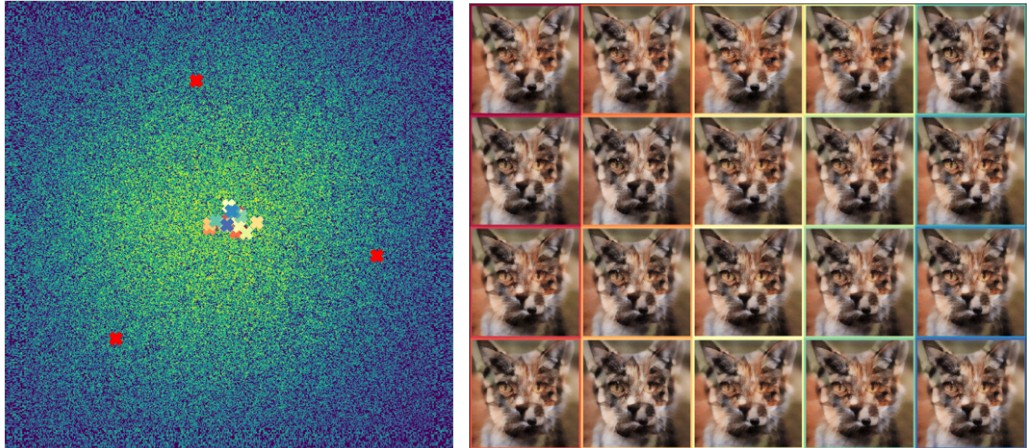

Figure 29: Decoded images (right) using 20 latents (left) from the 2D subspace, with highest $\delta$. Each image bounding box (right) is color coded according to the corresponding latent vector (left).

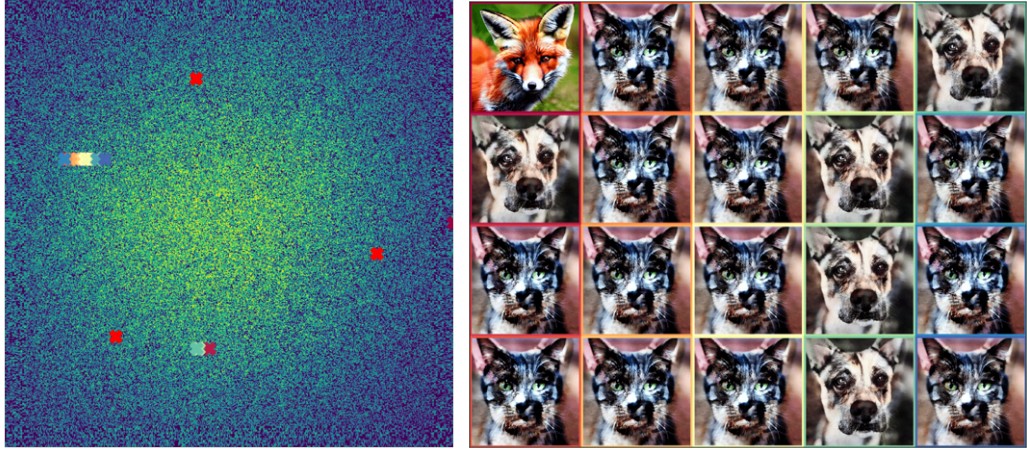

Figure 30: Decoded images (right) using 20 latents (left) from the 2D subspace, with lowest $\delta$. Each image bounding box (right) is color coded according to the corresponding latent vector (left).

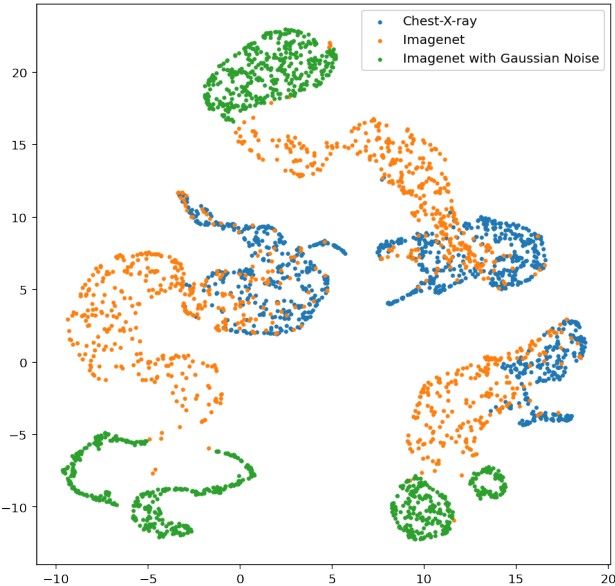

Figure 31: **UMAP visualization of the aggregated local geometry descriptors (local smoothness, local rank, and local complexity).** This reveals distinct, non-overlapping clusters, clearly separating the Imagenet, Imagenet Corrupted with Gaussian Noise, and Chest X-ray datasets. This visual evidence underscores the effectiveness of aggregating the descriptors to capture unique patterns within each dataset, demonstrating its ability to provide a meaningful and interpretable representation of the underlying data

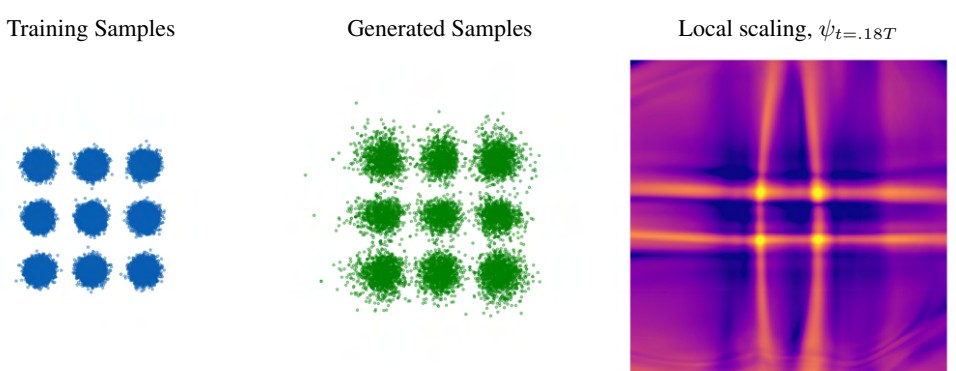

Figure 32: **Highest local scaling level sets are anti-modes.** In all of our experiments, we have observed that samples from the highest local scaling level sets have lower vendi score. We repeat the experiments from Fig. 15 for a mixture of nine gaussians. We see that the regions between the gaussian modes inside the domain of the data, i.e., the anti-modes, have higher local scaling, with the highest local scaling values at the center of the anti-modes.

