# OpenReview forum: "What Secrets Do Your Manifolds Hold? Understanding the Local Geometry of Generative Models"
_ICLR.cc/2025/Conference — ICLR 2025 Poster_

### Official Review · Reviewer_dk5T · 2024-11-03

**Soundness:** 3
**Presentation:** 3
**Contribution:** 4
**Rating:** 8
**Confidence:** 3

**Summary:**

Building on the continuous piecewise-linear hypothesis, this paper introduces three local geometric descriptors for generative models at specific stages of generation: scaling, rank, and smoothness. Using these descriptors, the paper empirically examines the local geometry of both a toy dataset and Stable Diffusion, drawing several conclusions about the learned image manifold. Finally, leveraging these descriptors, the paper proposes a novel reward model that guides the generation in Stable Diffusion, allowing control over specific local geometric properties.

**Strengths:**

This paper represents a valuable empirical contribution to understanding the local geometric properties of the image manifold learned by diffusion models. Based on the hypothesis that the output space of neural networks exhibits a continuous piecewise-linear structure, the learned data manifold can be approximated as a union of affine transformations across local regions. This foundation makes the proposed local geometric descriptors both intuitive and well-motivated. In empirical experiments, these descriptors align closely with human visual evaluation, underscoring their effectiveness. I believe this paper will significantly impact the study of image manifold geometry.

**Weaknesses:**

1. This paper lacks discussion with related works [1, 2, 3, 4]. The local dimensionality of the diffusion model-based image manifold has also been studied in those papers. However, as concurrent work, this won't significantly weaken the contribution of this paper. Better discussion about the relation and differences with those works could highlight the contribution of this paper.
2. The proposed method is not very strong. The authors provide only qualitative results without any quantitative evaluation or comparison to related works.
3. Some parts of the writing are unclear. I suggest including the numerical method used to approximate the calculation of local complexity (Definition 3) in this paper. Although this method was introduced in previous work, including it here would make the paper more complete and easier to understand.






[1] Stanczuk, Jan Pawel, Georgios Batzolis, Teo Deveney, and Carola-Bibiane Schönlieb. "Diffusion Models Encode the Intrinsic Dimension of Data Manifolds." In Forty-first International Conference on Machine Learning.

[2] Kamkari, Hamidreza, Brendan Leigh Ross, Rasa Hosseinzadeh, Jesse C. Cresswell, and Gabriel Loaiza-Ganem. "A Geometric View of Data Complexity: Efficient Local Intrinsic Dimension Estimation with Diffusion Models." arXiv preprint arXiv:2406.03537 (2024).

[3] Wang, Peng, Huijie Zhang, Zekai Zhang, Siyi Chen, Yi Ma, and Qing Qu. "Diffusion models learn low-dimensional distributions via subspace clustering." arXiv preprint arXiv:2409.02426 (2024).

[4] Chen, Siyi, Huijie Zhang, Minzhe Guo, Yifu Lu, Peng Wang, and Qing Qu. "Exploring low-dimensional subspaces in diffusion models for controllable image editing." arXiv preprint arXiv:2409.02374 (2024).

**Questions:**

1. In line 115, the paper discusses unrolling the sampling process of the diffusion model. Does this mean that when calculating the descriptor at a large timestep $t$, you need to compute the Jacobian iteratively over the neural network multiple times because the sampling steps are large? From my experiments, computing the Jacobian over the UNet in Stable Diffusion takes hours, even without unrolling. What is the computational cost of your implementation?

2. Are these three proposed descriptors redundant? From Figures 5, 12, and 13, it appears that as local scaling, local rank, and local complexity increase, the visual complexity of the images all increases. Seems we only need one descriptor for the image visual complexity. What specific, distinct properties do each of these three descriptors capture?

---

> ### Author Response · Authors · 2024-11-26
> **Author Response 1**
>
> We thank the reviewer for their constructive comments and suggestions. We are delighted to see that the reviewer has found our empirical contributions valuable, the local geometric descriptors intuitive and well-motivated, and for finding our work impactful. We answer specific comments/questions by the reviewer below:
>
> ------------
> ## Literature review
> > This paper lacks discussion with related works [1, 2, 3, 4]. The local dimensionality of the diffusion model-based image manifold has also been studied in those papers. However, as concurrent work, this won't significantly weaken the contribution of this paper. Better discussion about the relation and differences with those works could highlight the contribution of this paper.
>
> We thank the reviewer for their suggestion and sharing the citations. We will add a related work section with the mentioned papers to appendix and additional papers we have found on related topics. We will also add a summary of the related works in the main paper. We present some discussion on related works here also drawing contrasts with our paper:
>
> **Local geometry pre-diffusion.** Early applications of the local geometry of generative models involved improving the generation performance and/or utility of generative models via geometry inspired methods. For example, in [1] the authors proposed regularizing the contraction of the local geometry to learn better representations in autoencoders trained on MNIST and CIFAR10. The regularization penalty is employed via the norm of the input-output jacobian in [1], is an upper bound for local scaling presented in our paper. In [2] the authors provided visualizations on the curvature of pre-trained VAE latent spaces and proposed using an auxiliary variance estimator neural network to regularize the latent space geometry during generation. In [3] the authors perform latent space statistical inference problems, e.g., maximum likelihood inference, by training a separate neural network to approximate the Riemannian metric. In [4] the authors proposed a novel latent space sampling distribution based on the latent space geometry that allows uniformly sampling the learned data manifold of continuous-piecewise affine generators. The authors showed downstream benefits with fairness and diversity for such latent space samplers. While most of these methods discuss pre-diffusion architectures, their results are early demonstrations of how the local geometry can affect downstream generation. [3,4] also employ auxiliary Neural Networks to model an intrinsic property of a pre-trained generator, similar to how we propose using a reward model for Stable Diffusion.
>
> [1] Rifai et al., Contractive Auto-Encoders: Explicit Invariance During Feature Extraction, ICML 2011
>
> [2] Arvanitidis et al., LATENT SPACE ODDITY: ON THE CURVATURE OF DEEP GENERATIVE MODELS, ICLR 2018
>
> [3] Kuhnel et al., Latent Space Non-Linear Statistics, Arxiv 2018.
>
> [4] Humayun et al., MaGNET: Uniform Sampling from Deep Generative Network Manifolds Without Retraining, ICLR 2022

---

> ### Author Response · Authors · 2024-11-26
> **Author Response 1 Continued**
>
> **Local intrinsic dimensionality of diffusion models.** The local geometry of diffusion models and possible applications have garnered significant interest in recent years. In [6] the authors propose a method to compute the intrinsic dimensionality of diffusion models using the assumption that the score field is perpendicular to the data manifold. For any vector $x$ on the data manifold, the method requires computing the dimensionality of the score field around $x$ and subtracting it from the ambient dimension. To do that, the authors perform one step of the forward diffusion process $k$  times for $x$, denoise the $k$ noisy samples using the diffusion model and compute the rank of the data matrix containing denoised samples to obtain the intrinsic dimensionality. Compared to this method, we compute the dimensionality directly via a random estimation of the input-output jacobian SVD. We do not require any assumption on the score function vector field being perpendicular to the data manifold, which may not hold for a diffusion model that is not optimally trained or highly complex training datasets like LAION.
>
> In [5] the authors compute rank using the method proposed in [6] and show that local intrinsic dimensionality can be used for out-of-distribution (OOD) detection. This is analogous to our analysis in Sec 3 on the local geometry on or off the manifold. We can see that the intuition authors provided in [5] for diffusion models trained on smaller models and datasets e.g., FMNIST, MNIST, translate to larger scale models like Stable Diffusion trained on LAION as we have presented Fig 4, Fig 6 and Sec 4. Especially in Fig 6, we show that creating OOD samples with corruptions on Imagenet data (in-distribution), we can have an increase or decrease in negative-log likelihood (estimated via local scaling), with decrease for blurring corruptions and increase in noising corruptions.
>
> Concurrent work [7] has also shown the relationship between the intrinsic dimensionality (local rank) of Stable Diffusion scale models and the texture/visual complexity of generated images. We believe our analysis is much more holistic with three different geometric properties being measured compared to only local dimensinality. We i) show quantitatively how diversity measured via vendi score is higher for higher local scaling and rank values (Fig 8.). We have explored how rank and scaling evolves continuously across the latent space in Fig 4. We have presented how the geometry distribution varies as we continually perturb images via noise or blurring operations Fig 6. And finally in Sec 5 we have presented a method to guide generation using the local geometry to obtain downstream generation benefits.
>
> [5] Kamkari et al., A Geometric Explanation of the Likelihood OOD Detection Paradox, ICML 2024
> [6] Stanczuk et al., Diffusion Models Encode the Intrinsic Dimension of Data Manifolds, ICML 24
> [7] Kamkari et al., A Geometric View of Data Complexity: Efficient Local Intrinsic Dimension Estimation with Diffusion Models, NeurIPS 2024
>
> **Low dimensional subspaces in Diffusion Models.** In [8] and [9] empirically and theoretically the authors show that the rank of a diffusion model’s input-output jacobian is low dimensional and eigenvectors of the jacobian correspond to semantically meaningful directions in the latent space. This is reminiscent of past work on StyleGAN scale generative models [10].
> Results from [8,9] demonstrate that the CPWL assumption in our paper holds for a large class of diffusion models, and allows generation for diffusion models with attribute control. The result from [8] provide further insights on our empirical results. For example, in Fig. 4., rank visualization, we see that there exist parts of the latent space (outside the convex hull of the three anchor vectors) that have higher jacobian rank. The reason would be because the intrinsic dimensionality of the data is undefined off the data manifold and according to [8] the jacobian rank is upper bounded by the intrinsic dimensionality of the data.
>
> [8] Chen et al., Exploring Low-Dimensional Subspaces in Diffusion Models for Controllable Image Editing, ArXiv 2024
>
> [9] Wang et al., Diffusion models learn low-dimensional distributions via subspace clustering, ArXiv 2024
>
> [10] Balakrishnan et al., Rayleigh EigenDirections (REDs): GAN latent space traversals for multidimensional features, ArXiv 2022.

---

> ### Author Response · Authors · 2024-11-26
> **Author Response 1 Continued**
>
> **Misc.** Apart from the aforementioned works, [11] show that the emergence of generalization in diffusion models – when two networks separately trained on the same data learn the same mapping – can be attributed to the eigenspectrum and eigenvectors of the input-output jacobian. While we do not study the training dynamics of the local geometric descriptors in our paper, [11] suggests that the local geometry can be an important indicator of memorization and generalization emergence in diffusion models. In [12] the authors use the posterior principal components of a denoiser for uncertainty quantification. This work suggests that components with larger eigenvalues result in larger uncertainty which is directly related to the local scaling descriptors as it measures the product of non-zero singular values. While in [12] the authors propose using it for only a single image denoiser, we show that it generalizes for any diffusion model including Stable Diffusion scale text-to-image diffusion models.
>
> [11] Kadkhodaie et. al, Generalization in diffusion models arises from geometry-adaptive harmonic representations,  ICLR 2024
>
> [12] Manor et al., ON THE POSTERIOR DISTRIBUTION IN DENOISING: APPLICATION TO UNCERTAINTY QUANTIFICATION, ICLR 2024

---

> ### Author Response · Authors · 2024-11-26
> **Author Response 2**
>
> ----------
> ## Quantitative Evaluations of Reward Model
>
> > The proposed method is not very strong. The authors provide only qualitative results without any quantitative evaluation or comparison to related works.
>
>
> We agree with the reviewer and provide additional experiments to quantitatively evaluate the performance of our local scaling based reward model. We also use human preference RAHF [3] scores to evaluate the downstream generation before and after reward guidance.
>
> **Data preparation.** We obtain training data for the reward model by i) sampling $N$ images from Imagenet and encoding them to the Stable Diffusion latent space ii) adding noise using the forward diffusion process up to randomly chosen noise levels iii) for each latent computing the local scaling descriptor.
>
> To evaluate the performance of the reward model and dependency of the reward model on the number of training samples, we train multiple models for $N=${ $50K, 200K, 400K, 800K$ }. For evaluation, we generate $2560$ samples using the dreambooth live subject prompt templates [1], with Imagewoof [2] dogs as subjects that Stable Diffusion can generate with good variety. While Imagewoof dog classes are present in Imagenet therefore possibly in the training data, the dreambooth prompt templates contain a variety of settings that are not generally present in Imagenet, e.g., `a <subject> on top of pink fabric`.
>
> **Evaluation Setup.** We first sample Stable diffusion without any reward guidance and with classifier-free guidance of 7.5 to obtain baseline samples. We partition the range of local scaling values obtained for the baseline samples into $n=10$ bins (Rebuttal Fig 1 x-axis), where each bin contains images from a local scaling level set. Following that we use the same seed and prompts as the baseline samples to generate images using reward guidance to increase local scaling. For each bin or pre-guidance local scaling level set, we compare between the baseline samples and corresponding reward guided generations in the following three axes: i) change of local scaling ii) change of vendi (diversity) score iii) the change of human preference score (RAHF [3] aesthetic score).
>
> **Results.** In Rebuttal Fig.1-left below (anonymized link), we present the mean local scaling per bin with 95% confidence interval. Here the blue line represents the mean pre-guidance local scaling values, increasing from left to right. In Rebuttal Fig 1-middle and Rebuttal Fig-1 right, we present vendi scores and average predicted human preference scores [3]. For any bin, we present results for the reward guidance scale that maximizes the local scaling. We see that even for a model trained with $50K$ samples, we can have a considerable increase in the local scaling, diversity and aesthetic score for most of the bins. Changes in local scaling, vendi and aesthetic scores are higher for the lower pre-guidance local scaling level set bins compared to the higher pre-guidance local scaling level set bins.
>
> Rebuttal Fig 1. https://github.com/lukanon/rebuttal/blob/main/reward_guidance_effect.png
>
> We are adding the quantitative evaluation results to section 5 and will provide the updated manuscript shortly.
>
> [13] Ruiz et. al, DreamBooth: Fine Tuning Text-to-Image Diffusion Models for Subject-Driven Generation, CVPR 23.
>
> [14] https://github.com/fastai/imagenette
>
> [15] Liang et. al, Rich Human Feedback for Text-to-Image Generation, CVPR 24

---

> ### Author Response · Authors · 2024-11-26
> **Author Response 3**
>
> -----
> ## Clarifications about numerical methods
>
> > Some parts of the writing are unclear. I suggest including the numerical method used to approximate the calculation of local complexity (Definition 3) in this paper. Although this method was introduced in previous work, including it here would make the paper more complete and easier to understand.
>
> We apologize for the lack of clarity. We are adding the following implementation notes in Sec 2.2., “Extending beyond continuous piecewise-linear generators”. We are also adding in the appendix pseudocode for computing each of the descriptors for Stable Diffusion.
>
> **Implementation details for local scaling/rank:**  To avoid computing the singular values of the full input-output jacobian to obtain local scaling or rank – which will be significantly expensive for large networks – we obtain singular values via randomized SVD [4]. First we obtain a random projection matrix with orthonormal rows $\mathbf{W}$ with shape $k \times n$ such that $\mathbf{W}\mathbf{W}^T = \mathbb{I}_k$. Here $n$ is the dimensionality of the outputs generated by the network. We therefore approximate local scaling as:
>
> $\psi_\omega^{(trunc)} = \sum_{i=1}^k log(\sigma_i^{(trunc)})$, where $\sigma_i^{(trunc)}$ are the non-zero singular values of $\mathbf{W}\mathbf{A}_\omega$.
>
> For any $\omega$, if $\mathbf{W}$ forms a basis for the range of $\mathbf{A_\omega}$ then $\sigma_i  \approx \sigma_i^{(trunc)} \forall i={1,2…k}$ [4]. Therefore $\mathbf{WA_\omega}$ would provide us a low-rank approximation of $\mathbf{A_\omega}$.
>
> In our experiments we have tried two methods to obtain the projection matrix $W$
> 1) by obtaining the eigenvectors for the covariance matrix for a set of 50K randomly generated samples. This was suggested in [4].
> 2) by performing QR decomposition of a randomly initialized matrix. Implemented in jax.nn.initializers.orthogonal [5]
>
> We see that the performance difference between methods 1) and 2) are negligible therefore consider the cheaper alternative 2) and consider a fixed pre-computed $\mathbf{W}$ with k=120 for all $\mathbf{A_\omega}$ in our Stable Diffusion experiments.
>
> **Implementation details for local complexity:**  We will repeat the local complexity implementation details from [10] in the appendix. Here we provide a sketch for clarity.:
>
> 1) For a given latent vector $\mathbf{z}$, sample $k$ orthonormal vectors $\mathbf{v_i}$ and obtain set of vectors $V = $ { $\mathbf{z} \pm r\mathbf{v_i} \forall i=1,2,3...k$ } where $r$ is the radius of the local neighborhood to consider. $V$ is therefore the vertices of a cross-polytope in the latent space, centered on $\mathbf{z}$.
> 2) Forward pass all the vectors in $V$ though the network
> 3) For any layer in the network count the number of neurons for which there is a change of activation sign for any vector in $V$.
> 4) Sum the count over all layers to obtain local complexity. We can also sum over the final few layers since according to [10] the deeper layers non-linearities are more localized.
>
> For stable diffusion we use a $k=4$ and $r=0.0001$

---

> ### Author Response · Authors · 2024-11-26
> **Author Response 4**
>
> -----
> ## Reviewer Questions
>
> > In line 115, the paper discusses unrolling the sampling process of the diffusion model. Does this mean that when calculating the descriptor at a large timestep, you need to compute the Jacobian iteratively over the neural network multiple times because the sampling steps are large? From my experiments, computing the Jacobian over the UNet in Stable Diffusion takes hours, even without unrolling. What is the computational cost of your implementation?
>
> While we did mention in line 115 that the CPWL assumption holds for large networks, in practice we only compute the Jacobian for any single step for DDPM in Sec 3, or for the Stable Diffusion decoder network in most of our Stable Diffusion experiments.
>
> To avoid computing the singular values using the full input-output jacobian we use randomized SVD as we have mentioned in the previous response. The computation times for the local scaling and local rank (since both require one randomized SVD computation for one latent vector) ends up being 3929s for 1000 samples. For local complexity we require 113s for 1000 samples. All the estimates are for a JAX implementation of Stable Diffusion on TPUv3.
>
> Note that to train a reward model, we require the descriptors to be computed only once for each pre-trained model. If we compute the local scaling for 100k samples we require 173.1 TPU v3 hours which is equivalent to 54.58 V100 hours (according to Appendix A.3 [6]). Compared to 79,000 A100 hours required for Stable Diffusion training [7], ~24000 hours with enterprise level optimization [8], the computation required for the descriptors and reward model training is significantly small. The computation time for the local descriptors can be further reduced by using a smaller $k$ for our projection matrix $W$, or by using non-jacobian based methods, e.g., estimating the local scaling by measuring the change of volume for a unit norm l1-ball in the input space. We leave exploration of these directions for future work.
>
> *We will expand Section 2.2 “Extending Beyond Continuous Piecewise-Linear Generators” with all the aforementioned  details and add implementation pseudocode in the appendix. We thank the reviewer once again for the comment, we believe this is an important point that requires being highlighted in the paper to improve clarity and reproducibility*
>
> [4] Halko et. al, Finding Structure with Randomness: Probabilistic Algorithms for Constructing Approximate Matrix Decompositions, SIAM Review 2011.
>
> [5] https://jax.readthedocs.io/en/latest/_autosummary/jax.nn.initializers.orthogonal.html
>
> [6] Dhariwal et. al, Diffusion Models Beat GANs on Image Synthesis
>
> [7] https://www.mosaicml.com/blog/training-stable-diffusion-from-scratch-costs-160k
>
> [8] https://www.databricks.com/blog/stable-diffusion-2

---

> ### Author Response · Authors · 2024-11-26
> **Author Response 5**
>
> > Are these three proposed descriptors redundant? From Figures 5, 12, and 13, it appears that as local scaling, local rank, and local complexity increase, the visual complexity of the images all increases. Seems we only need one descriptor for the image visual complexity. What specific, distinct properties do each of these three descriptors capture?
>
> We thank the reviewer for asking this important clarifying question. In the qualitative Imagenet results presented in the Figures referenced by the reviewer, the difference between the descriptors is not as clear. Especially if we consider visual complexity, indeed local scaling and rank are both correlated with visual complexity in the qualitative results. While the descriptors are measuring distinct properties of the networks the are by definition correlated. Below we clarify the correlations between the descriptors and discuss how they are also capturing unique geometric aspects.
>
> **Local scaling** characterizes the change of volume by the affine slope $\mathbf{A_\omega}$ going from the latent space to the data manifold. **Local rank** characterizes the number of dimensions retained on the manifold after the network locally scales the latent space. Both local rank and scaling quantify first order properties of the CPWL operator. **Local complexity** approximates the ‘number of unique affine maps’ within a given neighborhood [10] by computing the number of CPWL knots intersecting an $\ell_1$ ball in the input/latent space. Therefore local complexity is a measure of ‘un-smoothness’ and quantifies local second-order properties of a CPWL operator.
>
> **Correlations between local scaling and rank.** By definition, local scaling and local rank are correlated, since both characterize the change of volume by the network input-output map at any input space linear region – also evident in Equations 2 and 4. This is also evident for our low dimensional DDPM setting presented in Fig 3., local rank and local scaling are highly correlated. The correlation persists throughout training as can be seen in Fig.3 right panel top and bottom. However in Fig 4., we can see that in the high-dimensional Stable Diffusion latent space, local scaling and rank are correlated but local rank has sharper changes spatially compared to local scaling.
>
> **Correlations between local complexity and rank.** There also exist correlations between local complexity and local rank due to the continuity of CPWL maps – between two neighboring linear regions $\omega_1$ and $\omega_2$, the corresponding slope matrices $\mathbf{A_{\omega_1}}$ and $\mathbf{A_{\omega_2}}$ differ by at most one row. Therefore between two neighboring regions the difference in local rank can be at most one.  Informally, the local rank in a neighborhood $V$ is lower bounded by the number of non-linearities in neighborhood $V$. This is evident in the empirical results presented in Fig 3 and Fig 4. In both figures, for input space neighborhoods with higher local complexity, we see a decrease in local rank. However, we do not observe sharp changes in local complexity as we observe in local rank in Fig 4. In Fig 3 we see that local rank is more discriminative of the data manifold compared to local complexity. Their training and denoising dynamics differ significantly as seen in Fig 3 rightmost column.
>
> **Qualitative and quantitative results on correlations.** We train a beta-VAE unconditionally on MNIST and present in Rebuttal Fig 5 samples from increasing local descriptor level sets from left to right along the columns. In rebuttal Fig 6, we present joint distributions of local scaling, complexity, rank and mean squared reconstruction error for training and test samples. We see that while local scaling, complexity and rank have some linear correlation, the classwise distribution is very different between the three (Rebuttal Fig 5).
>
> Rebuttal Fig 5. https://github.com/lukanon/rebuttal/blob/main/mnist_betavae_qualitative.png
>
> Rebuttal Fig 6. https://github.com/lukanon/rebuttal/blob/main/correlations.png
>
>
>
> [9] Balestriero et. al, A Spline Theory of Deep Learning, ICML 2018
>
> [10] Humayun et. al, Deep networks always grok and here is why, ICML 2024
>
> [11] Hanin et. al, Complexity of Linear Regions in Deep Networks, ICML 2019
>
> ---------
> ## Thanks
>
> We thank the reviewer for their detailed and constructive review which we believe has allowed us to significantly improve the paper. We kindly request the reviewer to ask any follow-up questions deemed necessary. We will update the manuscript adding the discussions above.

---

> > ### Comment · Reviewer_dk5T · 2024-11-29
> >
> > Thanks for your detailed answers. Before asking any further questions, I have raised my score. Because I believe this is a very strong empirical exploration paper. I have some further questions about the rebuttal Fig 1.
> >
> > 1. The reward model achieves the best result with only 50K training images. Is there any explanation for this?
> > 2. Why does the Vendi Score first increase and then decrease with the increase of the pre-guidance local scaling level set?

---

> > > ### Author Response · Authors · 2024-11-30
> > >
> > > We thank the reviewer for increasing the score and for their kind comments.
> > >
> > > 1.  As we had mentioned in Author Response 2, the reward model is trained on $($latent vector $,$ local scaling$)$ pairs where the latents are obtained from encoded random Imagenet samples. However when we evaluate the model, we use dreambooth live subject prompt templates to generate out-of imagenet domain samples. Using a larger training dataset size may possibly result in over-fitting on Imagenet which could be a reason for the relatively worse performance on our evaluation setup. We have also attempted training a model with $20K$ samples which under performs compared to models trained with $50K$ or above training samples. We believe there can be considerably better methods for generating training data for the reward model, e.g., by randomly generating samples using a dataset of diverse prompts. We leave further explorations on training the optimal geometry based reward model for future work.
> > >
> > >
> > > 2. We thank the reviewer for the important clarifying question. We observe that for the highest local scaling level sets, the vendi score (indicating diversity) drops compared to prior local scaling level sets. We have observed this phenomenon for Stable Diffusion, DiT-XL, beta-VAE trained on MNIST and toy settings as well. This happens due to the highest local scaling level sets converging to the least likely anti-modes of the output distribution. Since it is hard to qualitatively evaluate the collapse of diversity for the highest local scaling level sets in Stable Diffusion generations, we present here results for a toy setting for DDPM trained on a mixture of 2D gaussians and a beta-VAE trained on MNIST.
> > >
> > > **DDPM on 2D gaussian.** We train a DDPM on a mixture of nine gaussians as shown in Rebuttal Fig 8. The setting is identical to the experiments from Sec 3 involving a DDPM trained on a 2D dinosaur distribution. We observe in Rebuttal Fig. 8, that local scaling is higher in between the modes of the gaussian distributions and highest at the anti-modes.
> > >
> > > Rebuttal Fig 8 https://github.com/lukanon/rebuttal/blob/main/gaussian_mixture_ddpm.png
> > >
> > > **Beta-VAE trained on MNIST.** We train a beta-VAE on MNIST unconditionally, compute the local scaling values for training, test and generated images, and present in Rebuttal Fig. 9-right vendi scores for local scaling level sets increasing from left to right. We observe the same drop of vendi score in the highest local scaling level sets. Qualitatively in Rebuttal Fig 9-left, right most columns, we see that the highest local scaling level sets contain outlier images with low variance in the shape of the images, indicating a loss in diversity.
> > >
> > > Rebuttal Fig 9 https://github.com/lukanon/rebuttal/blob/main/betavae_mnist_vendi.png

---

### Official Review · Reviewer_KSLy · 2024-11-04

**Soundness:** 3
**Presentation:** 2
**Contribution:** 3
**Rating:** 6
**Confidence:** 3

**Summary:**

This paper studies the local geometry of the generative model and its relationship to the quality of the generation. Built upon the theory of continuous piecewise-linear generators, the authors propose to use three metrics: (1) local rank; (2) local scaling; and (3) local complexity to quantify the geometry of the generative model. Leveraging these descriptors, they try to understand how the local geometry relates to the generation process. Specifically, they show that local scaling is related to the complexity of the generated scenes, and can be used as a tool to detect whether the data is on the input (training) manifold. In addition, they propose to train a reward model to approximate the local scaling descriptor and present results showing more objects and complex backgrounds while increasing the value of the local scaling.

**Strengths:**

The idea is novel and the authors conduct extensive experiments in multiple scenarios. It is very interesting to understand the geometry of the model and its role in the generation process.

**Weaknesses:**

1. My biggest concern regarding this paper is the approachability to the audience. The three local geometric metrics discussed in the paper are very interesting. However, in terms of the method presentation, particularly the reward model built upon the local scaling descriptor, the current presentation lacks detailed descriptions, which makes me concerned about the reproducibility of the results.

2. The claim that "local geometry trajectories are discriminative of memorization" is relatively weak for the local complexity metric when the trend of the metrics is almost un-distinguishable when increasing the guidance scale.


Minors:
Line 323: I think it should be capitalized DDPM instead of "ddpm".

**Questions:**

1. Eqn.2: Should the singular values here be normalized?

2. Line 164: What's the definition of $S$, why the local scaling is proportional to the NLL of the generative model when the relationship between $\psi$ and $S$ is injective?

3. Lines 262-269: Where are the corresponding results and visualizations? Why do results at 0.17T show the minimized discrepancy between the on-manifold and off-manifold input space? And why does the minimized discrepancy indicate ``local geometry indicators have the highest distinction geometrically between on and off manifold vectors from the input space'' (should it be the opposite)?

4. Figure 3: What does $[-6, 6]^2$ mean? What's the x-axis of the plot on the right, is it the noise scale?

5. Figure 3 and Figure 4: What are the axes of the local generator? Does each pixel correspond to one input image? How is this being calculated?

6. Figure 4: What images correspond to the pixels in the center of the geometric desciptors?

7. Section 5: How is the reward function defined? Why the calculation of the Hessian matrix is required in the first place? The paper describes three local geometric descriptors, but why does the reward model only approximate the local scaling value?

---

> ### Author Response · Authors · 2024-11-26
> **Author Response 1**
>
> We would first like to thank the reviewer for their detailed review of our work and for their positive comments. We are delighted to see that the reviewer finds our idea novel, the three descriptors very interesting and our experiments extensive. We answer specific comments/questions by the reviewer below:
>
> --------------------
>
> ## Improving overall presentation, clarity and reproducibility
>
> > "My biggest concern regarding this paper is the approachability to the audience. The three local geometric metrics discussed in the paper are very interesting. However, in terms of the method presentation, particularly the reward model built upon the local scaling descriptor, the current presentation lacks detailed descriptions, which makes me concerned about the reproducibility of the results."
>
> We thank the reviewer for finding the three descriptors interesting, and apologize for the lack of clarity of the paper. To improve the overall clarity and reproducibility we are taking the following steps to revise the writing:
>
> ### **a) Adding more details on reward modeling and quantitative evaluation of the reward model**
>
> We agree with the reviewer and provide additional experiments to quantitatively evaluate the performance of our local scaling based reward model.
>
> **Data preparation.** We obtain training data for the reward model by i) sampling $N$ images from Imagenet and encoding them to the Stable Diffusion latent space ii) adding noise using the forward diffusion process up to randomly chosen noise levels iii) for each latent computing the local scaling descriptor.
>
> To evaluate the performance of the reward model and dependency of the reward model on the number of training samples, we train multiple models for $N=${ $50K, 200K, 400K, 800K$ }. For evaluation, we generate $2560$ samples using the dreambooth live subject prompt templates [1], with Imagewoof [2] dogs as subjects that Stable Diffusion can generate with good variety. While Imagewoof dog classes are present in Imagenet therefore possibly in the training data, the dreambooth prompt templates contain a variety of settings that are not generally present in Imagenet, e.g., `a <subject> on top of pink fabric`.
>
> **Evaluation Setup.** We first sample Stable diffusion without any reward guidance and with classifier-free guidance of 7.5 to obtain baseline samples. We partition the range of local scaling values obtained for the baseline samples into $n=10$ bins (Rebuttal Fig 1 x-axis), where each bin contains images from a local scaling level set. Following that we use the same seed and prompts as the baseline samples to generate images using reward guidance to increase local scaling. For each bin or pre-guidance local scaling level set, we compare between the baseline samples and corresponding reward guided generations in the following three axes: i) change of local scaling ii) change of vendi (diversity) score iii) the change of human preference score (RAHF [3] aesthetic score).
>
> **Results.** In Rebuttal Fig.1-left below (anonymized link), we present the mean local scaling per bin with 95% confidence interval. Here the blue line represents the mean pre-guidance local scaling values, increasing from left to right. In Rebuttal Fig 1-middle and Rebuttal Fig-1 right, we present vendi scores and average predicted human preference scores [3]. For any bin, we present results for the reward guidance scale that maximizes the local scaling. We see that even for a model trained with $50K$ samples, we can have a considerable increase in the local scaling, diversity and aesthetic score for most of the bins. Changes in local scaling, vendi and aesthetic scores are higher for the lower pre-guidance local scaling level set bins compared to the higher pre-guidance local scaling level set bins.
>
> Rebuttal Fig 1. https://github.com/lukanon/rebuttal/blob/main/reward_guidance_effect.png
>
> We are adding the quantitative evaluation results to section 5 and will provide the updated manuscript shortly.
>
> [1] Ruiz et. al, DreamBooth: Fine Tuning Text-to-Image Diffusion Models for Subject-Driven Generation, CVPR 23.
>
> [2] https://github.com/fastai/imagenette
>
> [3] Liang et. al, Rich Human Feedback for Text-to-Image Generation, CVPR 24

---

> ### Author Response · Authors · 2024-11-26
> **Author Response 1 Continued**
>
> ### **b) Adding implementation details in Sec 2.2 for Stable Diffusion. Adding pseudocode to appendix**
>
> As R1 had pointed out, there lacked clarity on how the local descriptors are computed for large networks like stable diffusion. We are adding the following implementation notes in Sec 2.2., “Extending beyond continuous piecewise-linear generators”. We are also adding pseudocode for computing each of the descriptors for Stable Diffusion.
>
> To avoid computing the singular values using the full input-output jacobian – which will be significantly expensive for large networks – for local scaling/rank computation we obtain singular values via randomized SVD [4]. First we obtain a random projection matrix with orthonormal rows $\mathbf{W}$ with shape $k \times n$ such that $\mathbf{W}\mathbf{W}^T = \mathbb{I}_k$. Here $n$ is the dimensionality of the outputs generated by the network. We therefore approximate local scaling as:
>
> $\psi_\omega^{(trunc)} = \sum_{i=1}^k log(\sigma_i^{(trunc)})$, where $\sigma_i^{(trunc)}$ are the non-zero singular values of $\mathbf{W}\mathbf{A}_\omega$.
>
> For any $\omega$, if $\mathbf{W}$ forms a basis for the range of $\mathbf{A_\omega}$ then $\sigma_i  \approx \sigma_i^{(trunc)} \forall i={1,2…k}$ [4]. Therefore $\mathbf{WA_\omega}$ would provide us a low-rank approximation of $\mathbf{A_\omega}$.
>
> In our experiments we have tried two methods to obtain the projection matrix $W$
> 1) by obtaining the eigenvectors for the covariance matrix for a set of 50K randomly generated samples. This was suggested in [4].
> 2) by performing QR decomposition of a randomly initialized matrix. Implemented in jax.nn.initializers.orthogonal [5]
>
> We see that the performance difference between methods 1) and 2) are negligible therefore consider the cheaper alternative 2) and consider a fixed pre-computed $\mathbf{W}$ with k=120 for all $\mathbf{A_\omega}$ in our Stable Diffusion experiments.
>
> [4] Halko et. al, Finding Structure with Randomness: Probabilistic Algorithms for Constructing Approximate Matrix Decompositions, SIAM Review 2011.
>
> [5] https://jax.readthedocs.io/en/latest/_autosummary/jax.nn.initializers.orthogonal.html
>
> ### **c) Adding discussions on the computation time required for each descriptor in the Appendix**
>
> The computation times for the local scaling and local rank computation (since both require one randomized SVD computation for one latent vector) ends up being 3929s for 1000 samples. For local complexity we require 113s for 1000 samples. All the estimates are for a JAX implementation of Stable Diffusion on TPUv3.
>
> Note that to train a reward model, we require the descriptors to be computed only once for each pre-trained model. If we compute the local scaling for 100k samples we require 173.1 TPU v3 hours which is equivalent to 54.58 V100 hours (according to Appendix A.3 [6]). Compared to 79,000 A100 hours required for Stable Diffusion training [7], ~24000 hours with enterprise level optimization [8], the computation required for the descriptors and reward model training is significantly small. The computation time for the local descriptors can be further reduced by using a smaller $k$ for our projection matrix $W$, or by using non-jacobian based methods, e.g., estimating the local scaling by measuring the change of volume for a unit norm l1-ball in the input space. We leave exploration of these directions for future work.
>
> *We will expand Section 2.2 “Extending Beyond Continuous Piecewise-Linear Generators” with all the aforementioned  details and add implementation pseudocode in the appendix. We thank the reviewer once again for the comment, we believe this is an important point that requires being highlighted in the paper to improve clarity and reproducibility. We are also doing multiple revisions and proofreading to reduce redundancy, moving Fig 3, to appendix and keeping columns 3 and 5 in main, making Fig 4, single column to increase space in main. We will shortly update the submission with our revised manuscript with the aforementioned changes.*
>
>
>
>
> [6] Dhariwal et. al, Diffusion Models Beat GANs on Image Synthesis
>
> [7] https://www.mosaicml.com/blog/training-stable-diffusion-from-scratch-costs-160k
>
> [8] https://www.databricks.com/blog/stable-diffusion-2

---

> ### Author Response · Authors · 2024-11-26
> **Author Response 2**
>
> ------------
>
> ## Memorization discrimination via local complexity.
>
> > The claim that "local geometry trajectories are discriminative of memorization" is relatively weak for the local complexity metric when the trend of the metrics is almost un-distinguishable when increasing the guidance scale.
>
> For the local complexity metric, the difference between the local complexity distribution of memorized and non-memorized prompts is not as significant as the local scaling distribution or the local rank distributions. At denoising step $50$ in Figure 8, the local complexity distribution between memorized and coco prompts have a t-test statistical significance (p-value) of $0.0004$ for a classifier free guidance scale of 7.5. We agree with the reviewer that increasing the classifier free guidance scale for COCO prompts to 9 would decrease the statistical significance for the local complexity descriptor making it non-discriminative. However for the local rank and local scaling distributions, the significance is still retained with a p-value of $< 0.000001$. In this case we will revise the claim and clarify that the significance might not hold for local complexity $\delta$ at higher classifier free guidance scales, while the local scaling and local rank descriptors are still discriminative.
>
> ------------
>
> ## Reviewer Questions
>
> > Eqn.2: Should the singular values here be normalized?
>
> We use the smooth rank [1] method of estimating the local rank which takes the exp of the shannon entropy of the singular value distribution. Smooth rank is frequently used to estimate rank given singular values of a matrix, since it doesn’t require determining a threshold to resolve non-zero singular value rounding errors. The normalization step is therefore required to compute the shannon entropy in Eqn 2.
>
> [1] Roy et. al, The effective rank: A measure of effective dimensionality. European signal processing conference 2007.
>
> > Line 164: What's the definition of S, why the local scaling is proportional to the NLL of the generative model when the relationship between  ψ  and S  is injective?
>
> We apologize for the lack of clarity here. Suppose we have an injective $\mathcal{G}: Z \rightarrow X$ mapping learned by a CPWL generator $\mathcal{G}$. Any linear region $\omega$ in the latent space CPWL partition $\Omega$ is mapped to a unique region on the output manifold. We define $S$ as:
>
> $S =$ { $\mathcal{G}(\mathbf{z}) \forall \mathbf{z} \in \omega $ } $=$ { $\mathbf{A_\omega}$$\mathbf{z}$ +$\mathbf{b_\omega}$ $\forall \mathbf{z} \in \omega $ }
>
> The change of volume from $\omega \rightarrow S$ is $\sqrt{det(\mathbf{A_\omega}^T \mathbf{A_\omega})}$. Therefore for a given latent $z \in \omega$ and output $x = \mathcal{G}(\mathbf{z})$:
>
> $p_{\mathcal{G}}(\mathbf{x}) = \frac{p_Z(\mathbf{z})}{\sqrt{det(\mathbf{A_\omega}^T \mathbf{A_\omega})}}$
>
> For a uniform latent distribution $p_Z(\mathbf{z}) =$ constant. Therefore,
>
> $p_{\mathcal{G}}(\mathbf{x}) \propto \frac{1}{\sqrt{det(\mathbf{A_\omega}^T \mathbf{A_\omega})}}$
>
> Taking negative log on both sides
>
> $-log(p_{\mathcal{G}}(\mathbf{x})) \propto log(\sqrt{det(\mathbf{A_\omega}^T \mathbf{A_\omega})})$
>
> Substituting definition of $\psi_\omega$ from Equation
>
> $-log(p_{\mathcal{G}}(\mathbf{x})) \propto \psi_\omega$
>
>
> Therefore for an injective piecewise linear map and uniform latent distribution, NLL on the output manifold $\propto \psi_\omega$. For a normal distribution in high-dimensional latent spaces, the proportionality can be assumed valid due to the concentration of measure on a thin shell.
>
>
> > Lines 262-269: Where are the corresponding results and visualizations?
>
> We apologize for the improper referencing here. We are referring to the results presented in Fig 3. We will re-write the section cross referencing the appropriate figures/sub-figures to increase clarity.

---

> ### Author Response · Authors · 2024-11-26
> **Author Response 3**
>
> > Why do results at 0.17T show the minimized discrepancy between the on-manifold and off-manifold input space? And why does the minimized discrepancy indicate ``local geometry indicators have the highest distinction geometrically between on and off manifold vectors from the input space'' (should it be the opposite)?
>
> We see that at $t \approx 0.17T$, for a DDPM trained for $125K$ optimization steps, the signed difference between on an off manifold local descriptors $E_{\mathcal{M}}[\Phi]$  $- E_{\bar{\mathcal{M}}}[\Phi]$ is minimized for local scaling and local rank, i.e., $\Phi \in $ {$\psi^t,  \nu^t \$} and is maximized for local complexity, i.e., $\Phi = \delta^t$. This can be observed in Fig 3 rightmost column for local scaling, complexity and rank. Note that we do not present the absolute difference, rather the signed difference between the on manifold estimate of local descriptor and off manifold estimate of local descriptor. We present the signed distance because this indicates that at $t \approx 0.17T$, local scaling and rank on the manifold is ‘lower’ than off the manifold. This is intuitive for local scaling since the on manifold negative log likelihood should be lower than off manifold negative log likelihood. The local rank being lower on the manifold is also reminiscent of the rank deficiency [2] property observed in deep neural network training, when neural networks reduce the rank of their weights monotonically. Note that for any network local rank is upper bounded by the minimum rank of all the weight matrices.
>
> [2] Feng et. al, Rank Diminishing in Deep Neural Networks, NeurIPS 2022.
>
> > Figure 3: What does [-6,6]^2 mean? What's the x-axis of the plot on the right, is it the noise scale?
>
> By [-6,6]^2 we denote the 2D domain in the input space of the DDPM for which we computed the local scaling values. X-axis in Fig 3 rightmost column is the noise scale, decreasing from left to right.
>
> > Figure 3 and Figure 4: What are the axes of the local generator? Does each pixel correspond to one input image? How is this being calculated?
>
> In Figure 3, each pixel corresponds to a 2D vector input to the DDPM. The axes in the images present in Figure 3 correspond to the two dimensions of the DDPM input space. In Figure 4., each pixel corresponds to a 64x64x4 dimensional latent vector in the Stable Diffusion latent space. First we randomly generate three latents with prompts “a fox”, “a dog” and “a cat”. We use the three generated latents to obtain two orthogonal vectors via Gram-Schmidt orthogonalization. The orthogonal vectors span the 2D subspace containing the three random latents. We sample a 2D grid of $512 \times 512$ vectors from the subspace, and for each vector we compute local scaling, rank and complexity independently. In Figure 4., rightmost column, we present the decoded images for the three latents.
>
> > Figure 4: What images correspond to the pixels in the center of the geometric descriptors?
>
> Appendix Figure 22 and appendix Figure 26 contain images decoded from latents from the center of the 2D subspace in Figure 4. Images are bounded by colored boxes, corresponding to the color of the marker presented in the 2D subspace visualizations.

---

> ### Author Response · Authors · 2024-11-26
> **Author Response 4**
>
> > The paper describes three local geometric descriptors, but why does the reward model only approximate the local scaling value?
>
> We apologize for the lack of clarity on this point in the current version. There are three reasons for choosing local scaling for reward modeling:
>
> i) Theoretically, local scaling is proportional to NLL on the image of the generator as discussed above in answer to question 2.
>
> ii) There exists correlations between local scaling, rank, and complexity, which is empirically visible across our experiments. Local scaling characterizes the change of volume by the affine slope $\mathbf{A_omega}$ going from the latent space to the data manifold. Local rank characterizes the number of dimensions retained on the manifold after locally scaling. Both local rank and scaling quantify first order properties of the CPWL operator. Local complexity approximates the ‘number of unique affine maps’ within a given neighborhood by computing the number of CPWL knots intersecting an $\ell_1$-ball in the input/latent space. By definition, local scaling, local rank and local complexity are correlated – scaling and rank both characterize the change of volume by the network input-output map and the local rank is lower bounded by local complexity. However, across our experiments in Section 4., we see that local scaling is most discriminative between qualitative aspects of downstream generation, e.g., memorization, aesthetic quality, diversity. Therefore, we choose local scaling as the candidate reward to showcase geometry based reward guidance.
>
> iii) Based on experimental results prior to Sec. 5., it was evident that local scaling has the potential to increase diversity, making it one of the first reward guidance methods to increase generation diversity. For example, in rebuttal Fig. 3., we can see that higher local scaling level sets have higher diversity. Note that the only prior work that proposed increasing diversity via a reward model was presented in [3].
>
> Rebuttal Fig. 3. Local scaling level set wise average vendi (top) and RAHF[3] human preference (bottom) scores, for 50K real images (black line) or images generated via Imagenet prompts (colored lines). We observe that higher local scaling level sets have higher vendi score indicating higher diversity in images.
> https://github.com/lukanon/rebuttal/blob/main/local_scaling_levelsets_vendi_humanpref.png
>
> [3] Hemmat et. al, Improving Geo-diversity of Generated Images with Contextualized Vendi Score Guidance, ECCV 2024.
>
> > Why the calculation of the Hessian matrix is required in the first place?
>
> In Sec 5., we consider local scaling as the target reward to be maximized via reward guidance. Since local scaling is a first-order measure that requires computing the jacobian of the input-output mapping by the generator, reward guidance via local scaling gradients would require computing the Hessian of the input-output mapping which can be computationally very expensive. To avoid computing the Hessian we train a reward model as a proxy and use the reward model gradients directly.
>
> ----------
>
> ## Thanks
>
> We thank the reviewer for their detailed and constructive review which we believe has allowed us to significantly improve the paper. We kindly request the reviewer to ask any follow-up questions deemed necessary. We will update the manuscript adding the discussions above.

---

> > ### Comment · Reviewer_KSLy · 2024-12-01
> >
> > I thank the authors for their reply. I appreciate the implementation details provided by the authors. I will keep my current score.

---

### Official Review · Reviewer_fW1y · 2024-11-04

**Soundness:** 2
**Presentation:** 2
**Contribution:** 3
**Rating:** 6
**Confidence:** 4

**Summary:**

This paper aims to study the local geometry of pretrained diffusion models and explore how different local geometry indicates different properties. It proposes three descriptors scaling (ψ), rank (ν), and smoothness (δ) to characterize the local geometry in diffusion models based on the continuous piecewise-linear (CPWL) assumption. The paper uses these descriptors to study the local geometry in classical DDPM and stable diffusion models. The paper conducts various experiments to show the following statements: 1. Local geometry of off manifold is different from on manifold. 2. Increasing local scaling leads to the generation of more complex image content. 3. Local geometry of Stable Diffusion is sensitive to image corruptions. 4. Local geometry of Stable Diffusion is sensitive to memorization of text prompts. Lastly, the paper proposes to guide generation with gradients of the descriptors (specifically local scaling). To avoid heavy computation of Hessian, they actually train a reward model to approximate local scaling. With this geometric guidance, the paper can control content complexity in the generated images.

**Strengths:**

1. The paper is studying the local geometry from a comprehensive perspective, modeling scaling (ψ), rank (ν), and smoothness (δ).

2. The paper shows experiment results from several perspectives including off and on manifold, content complexity, image corruption, and text prompt memorization.

3. The paper utilizes local scaling to control the generated image complexity.

**Weaknesses:**

1. Lack of literature review. There exists prior work [1] related to out-of-domain detection with local geometry, and [2] related to uncertainty quantification with local geometry. Even though the metrics are not exactly the same, it is worth discussing these previous works, as well as other ones related to the manifold of diffusion models.

2. Lack of clarity and technical details. See the questions part.

3. Lack of Justification of the continuous piecewise-linear (CPWL) assumption: The paper does not provide justification that CPWL can be a good approximation for DDPM and stable diffusion. In contrast, recent work [3] has experimentally verified such local linearity.

4. Redundancy of metrics: Though all three metrics are interesting, it seems rank (ν), and smoothness (δ) are redundant since most observations and applications can be correlated only with scaling.

[1] A Geometric Explanation of the Likelihood OOD Detection Paradox. Hamidreza Kamkari et al.

[2] On the Posterior Distribution in Denoising: Application to Uncertainty Quantification. Hila Manor et al.

[3] Exploring Low-Dimensional Subspaces in Diffusion Models for Controllable Image Editing. Siyi Chen et al.

**Questions:**

(a)  Where does equation (3) come from?

(b) In equation (7), how is B defined? (i. How to choose number of dimensions P, is it the same aound any latent $z$? ii. How to choose columns in B? iii. Why a constant projection B can be used for all neighbor around the specific $z$?)

(c) What explicitly are the input space vectors for DDPM and Stable Diffusions? Are they the inputs to the denoising UNet, or could they be features in the middle layers of the UNet?

(d) In section 3.1 L250, "input vectors within 0.05 units of the training data", how is 0.05 chosen and why it is a reasonable value?

(e) In Figure 4, how are the three anchors (representing latent from 3 text prompts) labeled in the latent space and what are other points?

---

> ### Author Response · Authors · 2024-11-26
> **Author Response 1**
>
> We thank the reviewer for their constructive comments and suggestions. We are delighted to see that the reviewer has found our study on geometric descriptors comprehensive. We answer specific comments/questions by the reviewer below:
>
> ------------
> ## Literature review
> > Lack of literature review. There exists prior work [1] related to out-of-domain  detection with local geometry, and [2] related to uncertainty quantification with local geometry. Even though the metrics are not exactly the same, it is worth discussing these previous works, as well as other ones related to the manifold of diffusion models.
>
> We thank the reviewer for their suggestion and sharing the citations. We will add a related work section with the mentioned papers to appendix and additional papers we have found on related topics. We will also add a summary of the related works in the main paper. We present some discussion on related works here:
>
> **Local geometry pre-diffusion.** Early applications of the local geometry of generative models involved improving the generation performance and/or utility of generative models via geometry inspired methods. For example, in [1] the authors proposed regularizing the contraction of the local geometry to learn better representations in autoencoders trained on MNIST and CIFAR10. The regularization penalty is employed via the norm of the input-output jacobian in [1], is an upper bound for local scaling presented in our paper. In [2] the authors provided visualizations on the curvature of pre-trained VAE latent spaces and proposed using an auxiliary variance estimator neural network to regularize the latent space geometry during generation. In [3] the authors perform latent space statistical inference problems, e.g., maximum likelihood inference, by training a separate neural network to approximate the Riemannian metric. In [4] the authors proposed a novel latent space sampling distribution based on the latent space geometry that allows uniformly sampling the learned data manifold of continuous-piecewise affine generators. The authors showed downstream benefits with fairness and diversity for such latent space samplers. While most of these methods discuss pre-diffusion architectures, their results are early demonstrations of how the local geometry can affect downstream generation. [2,3] also employ auxiliary Neural Networks to model an intrinsic property of a pre-trained generator, similar to how we propose using a reward model for Stable Diffusion.
>
> [1] Rifai et al., Contractive Auto-Encoders: Explicit Invariance During Feature Extraction, ICML 2011
>
> [2] Arvanitidis et al., LATENT SPACE ODDITY: ON THE CURVATURE OF DEEP GENERATIVE MODELS, ICLR 2018
>
> [3] Kuhnel et al., Latent Space Non-Linear Statistics, Arxiv 2018.
>
> [4] Humayun et al., MaGNET: Uniform Sampling from Deep Generative Network Manifolds Without Retraining, ICLR 2022

---

> ### Author Response · Authors · 2024-11-26
> **Author Response 1 Continued**
>
> **Local intrinsic dimensionality of diffusion models.** The local geometry of diffusion models and possible applications have garnered significant interest in recent years. In [6] the authors propose a method to compute the intrinsic dimensionality of diffusion models using the assumption that the score field is perpendicular to the data manifold. For any vector $x$ on the data manifold, the method requires computing the dimensionality of the score field around $x$ and subtracting it from the ambient dimension. To do that, the authors perform one step of the forward diffusion process $k$  times for $x$, denoise the $k$ noisy samples using the diffusion model and compute the rank of the data matrix containing denoised samples to obtain the intrinsic dimensionality. Compared to this method, we compute the dimensionality directly via a random estimation of the input-output jacobian SVD. We do not require any assumption on the score function vector field being perpendicular to the data manifold, which may not hold for a diffusion model that is not optimally trained or highly complex training datasets like LAION.
>
> In [5] the authors compute rank using the method proposed in [6] and show that local intrinsic dimensionality can be used for out-of-distribution (OOD) detection. This is analogous to our analysis in Sec 3 on the local geometry on or off the manifold. We can see that the intuition authors provided in [5] for diffusion models trained on smaller models and datasets e.g., FMNIST, MNIST, translate to larger scale models like Stable Diffusion trained on LAION as we have presented Fig 4, Fig 6 and Sec 4. Especially in Fig 6, we show that creating OOD samples with corruptions on Imagenet data (in-distribution), we can have an increase or decrease in negative-log likelihood (estimated via local scaling), with decrease for blurring corruptions and increase in noising corruptions.
>
> Concurrent work [7] has also shown the relationship between the intrinsic dimensionality (local rank) of Stable Diffusion scale models and the texture/visual complexity of generated images. We believe our analysis is much more holistic with three different geometric properties being measured compared to only local dimensinality. We i) show quantitatively how diversity measured via vendi score is higher for higher local scaling and rank values (Fig 8.). We have explored how rank and scaling evolves continuously across the latent space in Fig 4. We have presented how the geometry distribution varies as we continually perturb images via noise or blurring operations Fig 6. And finally in Sec 5 we have presented a method to guide generation using the local geometry to obtain downstream generation benefits.
>
> **Low dimensional subspaces in Diffusion Models.** In [8] and [9] empirically and theoretically the authors show that the rank of a diffusion model’s input-output jacobian is low dimensional and eigenvectors of the jacobian correspond to semantically meaningful directions in the latent space. This is reminiscent of past work on StyleGAN scale generative models [10].
> Results from [8,9] demonstrate that the CPWL assumption in our paper holds for a large class of diffusion models, and allows generation for diffusion models with attribute control. The result from [8] provide further insights on our empirical results. For example, in Fig. 4., rank visualization, we see that there exist parts of the latent space (outside the convex hull of the three anchor vectors) that have higher jacobian rank. The reason would be because the intrinsic dimensionality of the data is undefined off the data manifold and according to [8] the jacobian rank is upper bounded by the intrinsic dimensionality of the data.
>
>
> [5] Kamkari et al., A Geometric Explanation of the Likelihood OOD Detection Paradox, ICML 2024
>
> [6] Stanczuk et al., Diffusion Models Encode the Intrinsic Dimension of Data Manifolds, ICML 2024
>
> [7] Kamkari et al., A Geometric View of Data Complexity: Efficient Local Intrinsic Dimension Estimation with Diffusion Models, NeurIPS 2024
>
> [8] Chen et al., Exploring Low-Dimensional Subspaces in Diffusion Models for Controllable Image Editing, ArXiv 2024
>
> [9] Wang et al., Diffusion models learn low-dimensional distributions via subspace clustering, ArXiv 2024
>
> [10] Balakrishnan et al., Rayleigh EigenDirections (REDs): GAN latent space traversals for multidimensional features, ArXiv 2022.

---

> ### Author Response · Authors · 2024-11-26
> **Author Response 1 Continued**
>
> **Misc.** Apart from the aforementioned works, [11] show that the emergence of generalization in diffusion models – when two networks separately trained on the same data learn the same mapping – can be attributed to the eigenspectrum and eigenvectors of the input-output jacobian. While we do not study the training dynamics of the local geometric descriptors in our paper, [11] suggests that the local geometry can be an important indicator of memorization and generalization emergence in diffusion models. In [12] the authors use the posterior principal components of a denoiser for uncertainty quantification. This work suggests that components with larger eigenvalues result in larger uncertainty which is directly related to the local scaling descriptors as it measures the product of non-zero singular values. While in [12] the authors propose using it for only a single image denoiser, we show that it generalizes for any diffusion model including Stable Diffusion scale text-to-image diffusion models.
>
> [11] Kadkhodaie et. al, Generalization in diffusion models arises from geometry-adaptive harmonic representations,  ICLR 2024
>
> [12] Manor et al., ON THE POSTERIOR DISTRIBUTION IN DENOISING: APPLICATION TO UNCERTAINTY QUANTIFICATION, ICLR 2024

---

> ### Author Response · Authors · 2024-11-26
> **Author Response 2**
>
> --------
> ## Lack of clarity and technical details
>
> We apologize for the lack of clarity in our paper, apart from the specific questions asked by the reviewer, we will add the following details to make the paper more rigorous, increase readability and reproducibility.
>
>
> ### **a) Adding more details on reward modeling and quantitative evaluation of the reward model**
>
> We agree with the reviewer and provide additional experiments to quantitatively evaluate the performance of our local scaling based reward model.
>
> **Data preparation.** We obtain training data for the reward model by i) sampling $N$ images from Imagenet and encoding them to the Stable Diffusion latent space ii) adding noise using the forward diffusion process up to randomly chosen noise levels iii) for each latent computing the local scaling descriptor.
>
> To evaluate the performance of the reward model and dependency of the reward model on the number of training samples, we train multiple models for $N=${ $50K, 200K, 400K, 800K$ }. For evaluation, we generate $2560$ samples using the dreambooth live subject prompt templates [1], with Imagewoof [2] dogs as subjects that Stable Diffusion can generate with good variety. While Imagewoof dog classes are present in Imagenet therefore possibly in the training data, the dreambooth prompt templates contain a variety of settings that are not generally present in Imagenet, e.g., `a <subject> on top of pink fabric`.
>
> **Evaluation Setup.** We first sample Stable diffusion without any reward guidance and with classifier-free guidance of 7.5 to obtain baseline samples. We partition the range of local scaling values obtained for the baseline samples into $n=10$ bins (Rebuttal Fig 1 x-axis), where each bin contains images from a local scaling level set. Following that we use the same seed and prompts as the baseline samples to generate images using reward guidance to increase local scaling. For each bin or pre-guidance local scaling level set, we compare between the baseline samples and corresponding reward guided generations in the following three axes: i) change of local scaling ii) change of vendi (diversity) score iii) the change of human preference score (RAHF [3] aesthetic score).
>
> **Results.** In Rebuttal Fig.1-left below (anonymized link), we present the mean local scaling per bin with 95% confidence interval. Here the blue line represents the mean pre-guidance local scaling values, increasing from left to right. In Rebuttal Fig 1-middle and Rebuttal Fig-1 right, we present vendi scores and average predicted human preference scores [3]. For any bin, we present results for the reward guidance scale that maximizes the local scaling. We see that even for a model trained with $50K$ samples, we can have a considerable increase in the local scaling, diversity and aesthetic score for most of the bins. Changes in local scaling, vendi and aesthetic scores are higher for the lower pre-guidance local scaling level set bins compared to the higher pre-guidance local scaling level set bins.
>
> Rebuttal Fig 1. https://github.com/lukanon/rebuttal/blob/main/reward_guidance_effect.png
>
> We are adding the quantitative evaluation results to section 5 and will provide the updated manuscript shortly.
>
> [1] Ruiz et. al, DreamBooth: Fine Tuning Text-to-Image Diffusion Models for Subject-Driven Generation, CVPR 23.
>
> [2] https://github.com/fastai/imagenette
>
> [3] Liang et. al, Rich Human Feedback for Text-to-Image Generation, CVPR 24
>
> ### **b) Adding implementation details in Sec 2.2 for Stable Diffusion. Adding pseudocode to appendix**
>
> The initial submission lacked clarity on how the local descriptors are computed for large networks like stable diffusion. We are adding the following implementation notes in Sec 2.2., “Extending beyond continuous piecewise-linear generators”. We are also adding pseudocode for computing each of the descriptors for Stable Diffusion.
>
> To avoid computing the singular values using the full input-output jacobian – which will be significantly expensive for large networks – for local scaling/rank computation we obtain singular values via randomized SVD [4]. First we obtain a random projection matrix with orthonormal rows $\mathbf{W}$ with shape $k \times n$ such that $\mathbf{W}\mathbf{W}^T = \mathbb{I}_k$. Here $n$ is the dimensionality of the outputs generated by the network. We therefore approximate local scaling as:
>
> $\psi_\omega^{(trunc)} = \sum_{i=1}^k log(\sigma_i^{(trunc)})$, where $\sigma_i^{(trunc)}$ are the non-zero singular values of $\mathbf{W}\mathbf{A}_\omega$.
>
> For any $\omega$, if $\mathbf{W}$ forms a basis for the range of $\mathbf{A_\omega}$ then $\sigma_i  \approx \sigma_i^{(trunc)} \forall i={1,2…k}$ [4]. Therefore $\mathbf{WA_\omega}$ would provide us a low-rank approximation of $\mathbf{A_\omega}$.

---

> ### Author Response · Authors · 2024-11-26
> **Author Response 2 Continued**
>
> In our experiments we have tried two methods to obtain the projection matrix $W$
> 1) by obtaining the eigenvectors for the covariance matrix for a set of 50K randomly generated samples. This was suggested in [4].
> 2) by performing QR decomposition of a randomly initialized matrix. Implemented in jax.nn.initializers.orthogonal [5]
>
> We see that the performance difference between methods 1) and 2) are negligible therefore consider the cheaper alternative 2) and consider a fixed pre-computed $\mathbf{W}$ with k=120 for all $\mathbf{A_\omega}$ in our Stable Diffusion experiments.
>
> [4] Halko et. al, Finding Structure with Randomness: Probabilistic Algorithms for Constructing Approximate Matrix Decompositions, SIAM Review 2011.
>
> [5] https://jax.readthedocs.io/en/latest/_autosummary/jax.nn.initializers.orthogonal.html
>
>
> ### **c) Adding discussions on the computation time required for each descriptor in the Appendix**
>
> The computation times for the local scaling and local rank computation (since both require one randomized SVD computation for one latent vector) ends up being 3929s for 1000 samples. For local complexity we require 113s for 1000 samples. All the estimates are for a JAX implementation of Stable Diffusion on TPUv3.
>
> Note that to train a reward model, we require the descriptors to be computed only once for each pre-trained model. If we compute the local scaling for 100k samples we require 173.1 TPU v3 hours which is equivalent to 54.58 V100 hours (according to Appendix A.3 [6]). Compared to 79,000 A100 hours required for Stable Diffusion training [7], ~24000 hours with enterprise level optimization [8], the computation required for the descriptors and reward model training is significantly small. The computation time for the local descriptors can be further reduced by using a smaller $k$ for our projection matrix $W$, or by using non-jacobian based methods, e.g., estimating the local scaling by measuring the change of volume for a unit norm l1-ball in the input space. We leave exploration of these directions for future work.
>
> *We will expand Section 2.2 “Extending Beyond Continuous Piecewise-Linear Generators” with all the aforementioned  details and add implementation pseudocode in the appendix. We thank the reviewer once again for the comment, we believe this is an important point that requires being highlighted in the paper to improve clarity and reproducibility. We are also doing multiple revisions and proofreading to reduce redundancy, moving Fig 3, to appendix and keeping columns 3 and 5 in main, making Fig 4, single column to increase space in main. We will shortly update the submission with our revised manuscript with the aforementioned changes.*
>
>
>
>
> [6] Dhariwal et. al, Diffusion Models Beat GANs on Image Synthesis
>
> [7] https://www.mosaicml.com/blog/training-stable-diffusion-from-scratch-costs-160k
>
> [8] https://www.databricks.com/blog/stable-diffusion-2

---

> ### Author Response · Authors · 2024-11-26
> **Author Response 3**
>
> ## Justification  for CPWL assumption
>
> > The paper does not provide justification that CPWL can ... recent work [3] has experimentally verified such local linearity.
>
> **Why we start from a CPWL assumption.** One of the main benefits of the CPWL assumption is that for any arbitrary deep neural network as long as the non-linearities are continuous piecewise linear, e.g., ReLU, Leaky-ReLU, Triangle, MaxPool, Batch Normalization, only first-order and second-order (at the knots) derivatives are non-zero. That is why we use the three descriptors to characterize the network locally, where local scaling and rank measure first order local properties and local complexity measure second order properties – providing us a comprehensive summary of the local geometry.
>
> **GeLU Activation.** Stable Diffusion v1.4 and the DDPM in our experiments, employ the GeLU activation function making them non CPWL. For networks with such smooth activation functions or non-piecewise-linear non-linearities, our descriptors become first order Taylor approximations. Prior work has shown that smooth activations lead to a soft VQ partition [1] instead of a hard VQ partition like in CPWL networks. This entails that much of the local linear structure we expect in CPWL maps are retained even if we employ smooth approximations of ReLU, e.g., GeLU activations. We have mentioned this in Sec 2.2 “Extending beyond continuous piecewise-linear generators” and will highlight it further in the paper.
>
> **DDPM linearity.** One empirical result to verify how close to CPWL the learned function is for DDPM, is presented in Fig 3 2nd row, which shows the spatial distribution of local complexity in the input space. Here, every continuous line we see corresponds to the zero level set of one GeLU activated neuron in the network. For ReLU activated neurons, these lines would correspond to the knots of the CPWL partition and they would be continuous-piecewise linear. However, even with GeLU activated neurons for DDPM, in Fig 3 we observe that the neuron zero level sets are quite close to being piecewise-linear. This provides qualitative evidence that the CPWL assumption is not a strong assumption for our DDPM setting.
>
> **Empirical results on local linearity from [2].** We thank the reviewer for sharing the citation for [2] which we will cite in Sec 2 of our paper to support our CPWL assumption. In [2] the authors empirically demonstrate that the CPWL assumption holds for a large class of diffusion models, and allows generation for diffusion models with attribute control. In [2] the first-order Taylor approximation in Eq. 4 is analogous to Eq. 2 in our paper.
>
> Our results also provide further insight on the result from [2] that the singular vectors of the input-output Jacobian always lie in low-dimensional semantic subspaces. For example, in Fig. 4., rank visualization, we see that there exist parts of the latent space (outside the convex hull of the three anchor vectors) that have higher jacobian rank. The reason would be since the according to [2] the jacobian rank is upper bounded by the intrinsic dimensionality of the data but this is undefined for a diffusion model’s latent space locations where the data distribution does not have support. This could also be an indication of the sharp changes in rank in Fig 4., being due to the data distribution.
>
> **Additional results for DiT.** To show that the intuition that we get for CPWL networks translate to non-CPWL networks like DiTs, we perform additional experiments and analysis on a DiT-XL trained on Imagenet. For the DiT we compute the descriptors for the transformer network, conditioned on timestep t=0, i.e., at noise level zero. We generate 5120 images conditioned on Imagewoof [2] classes and present in Rebuttal Fig 2 below, increasing local scaling level sets from left to right. We see that similar to Fig 6 from the paper, DiT exhibits a qualitative correlation between visual complexity and local scaling. For additional analysis we repeat the Stable Diffusion experiments from Sec 4 (Figure 9) on the relation between diversity and local scaling for DiT. We see that similar to Stable Diffusion, for increasing local scaling level sets, the diversity of images increase and then drop for the highest local scaling level sets.
>
> Fig 2: Randomly sampled images from local scaling level sets for which Vendi scores are presented in Fig 2:
> https://github.com/lukanon/rebuttal/blob/main/imagewoof_combined_levelsets.jpg
>
> Fig 3: Vendi score for increasing local scaling level sets, computed for a DiT-XL Transformer network trained on Imagenet.
>  https://github.com/lukanon/rebuttal/blob/main/imagewoof_combined_vendi.pdf
>
> [1]  Balestriero et al., From Hard to Soft: Understanding Deep Network Nonlinearities via Vector Quantization and Statistical Inference, ICLR 2019
> [2] Chen et. al, Exploring Low-Dimensional Subspaces in Diffusion Models for Controllable Image Editing, NeurIPS 2024

---

> ### Author Response · Authors · 2024-11-26
> **Author Response 4**
>
> -----
> ## Redundancy  of metrics
>
> >  Though all three metrics are interesting, it seems rank (ν), and smoothness (δ) are redundant since most observations and applications can be correlated only with scaling.
>
> Neural networks with only CPWL non-linearities are CPWL operators, and their input-output mapping can be fully characterized by the regions in the input space partitioning and the affine operator per region (see Equation 1). This is irrespective of the dimensionality of the input space or latent space [9]. Given that, we can measure three characteristics of the CPWL mapping via the three descriptors. **Local scaling** characterizes the change of volume by the affine slope $\mathbf{A_\omega}$ going from the latent space to the data manifold. **Local rank** characterizes the number of dimensions retained on the manifold after the network locally scales the latent space. Both local rank and scaling quantify first order properties of the CPWL operator. Local complexity approximates the ‘number of unique affine maps’ within a given neighborhood [10] by computing the number of CPWL knots intersecting an $\ell_1$ ball in the input/latent space. Therefore local complexity is a measure of ‘un-smoothness’ and quantifies local second-order properties of a CPWL operator.
>
> **Correlations between local scaling and rank.** By definition, local scaling and local rank are correlated, since both characterize the change of volume by the network input-output map at any input space linear region – also evident in Equations 2 and 4. This is also evident for our low dimensional DDPM setting presented in Fig 3., local rank and local scaling are highly correlated. The correlation persists throughout training as can be seen in Fig.3 right panel top and bottom. However in Fig 4., we can see that in the high-dimensional Stable Diffusion latent space, local scaling and rank are correlated but local rank has sharper changes spatially compared to local scaling.
>
> **Correlations between local complexity and rank.** There also exist correlations between local complexity and local rank due to the continuity of CPWL maps – between two neighboring linear regions $\omega_1$ and $\omega_2$, the corresponding slope matrices $\mathbf{A_{\omega_1}}$ and $\mathbf{A_{\omega_2}}$ differ by at most one row. Therefore between two neighboring regions the difference in local rank can be at most one.  Informally, the local rank in a neighborhood $V$ is lower bounded by the number of non-linearities in neighborhood $V$. This is evident in the empirical results presented in Fig 3 and Fig 4. In both figures, for input space neighborhoods with higher local complexity, we see a decrease in local rank. However, we do not observe sharp changes in local complexity as we observe in local rank in Fig 4. In Fig 3 we see that local rank is more discriminative of the data manifold compared to local complexity. Their training and denoising dynamics differ significantly as seen in Fig 3 rightmost column.
>
> **Qualitative and quantitative results on correlations.** We train a beta-VAE unconditionally on MNIST and present in Rebuttal Fig 5 samples from increasing local descriptor level sets from left to right along the columns. In rebuttal Fig 6, we present joint distributions of local scaling, complexity, rank and mean squared reconstruction error for training and test samples. We see that while local scaling, complexity and rank have some linear correlation, the classwise distribution is very different between the three (Rebuttal Fig 5).
>
> Rebuttal Fig 5. https://github.com/lukanon/rebuttal/blob/main/mnist_betavae_qualitative.png
>
> Rebuttal Fig 6. https://github.com/lukanon/rebuttal/blob/main/correlations.png
>
>
>
> [9] Balestriero et. al, A Spline Theory of Deep Learning, ICML 2018
>
> [10] Humayun et. al, Deep networks always grok and here is why, ICML 2024
>
> [11] Hanin et. al, Complexity of Linear Regions in Deep Networks, ICML 2019

---

> ### Author Response · Authors · 2024-11-26
> **Author Response 5**
>
> -----
> ## Reviewer Questions
>
> > Where does equation (3) come from?
>
>
> We apologize for the lack of clarity here. Suppose we have an injective $\mathcal{G}: Z \rightarrow X$ mapping learned by a CPWL generator $\mathcal{G}$. Any linear region $\omega$ in the latent space CPWL partition $\Omega$ is mapped to a unique region on the output manifold. We define $S$ as:
>
> $S =$ { $\mathcal{G}(\mathbf{z}) \forall \mathbf{z} \in \omega $ } $=$ { $\mathbf{A_\omega}$$\mathbf{z}$ +$\mathbf{b_\omega}$ $\forall \mathbf{z} \in \omega $ }
>
> The change of volume from $\omega \rightarrow S$ is $\sqrt{det(\mathbf{A_\omega}^T \mathbf{A_\omega})}$. Therefore for any latent $z$ and output $x = \mathcal{G}(\mathbf{z})$:
>
> $p_{\mathcal{G}}(\mathbf{x}) = \sum_{\forall \omega \in \Omega} \frac{p_Z(\mathbf{z})}{\sqrt{det(\mathbf{A_\omega}^T \mathbf{A_\omega})}}$$\mathbb{1}_{z \in \omega}$
>
> For any $\mathbf{z_1} \in \omega_1$ the sum from the above equation can be ignored, since for all other regions the value would be zero.
>
> Taking negative log and expectation on both sides the conditional entropy becomes
>
> $H(p_{\mathcal{G}}(\mathbf{x_1}) ; \mathbf{z} \in \omega_1) = H(p_Z(\mathbf{z_1})) + log(\sqrt{det(\mathbf{A_{\omega_1}}^T \mathbf{A_{\omega_1}})})$
>
> For a uniform latent distribution and two regions $\omega_1$ and $\omega_2$, substituting the second term above with $\psi_{\omega_1}$
>
> $H(p_{\mathcal{G}}(\mathbf{x_1}) ; \mathbf{z_1} \in \omega_1)$   $-$   $H(p_{\mathcal{G}}(\mathbf{x_2}) ; \mathbf{z_2} \in \omega_2) = \psi_{\omega_1} - \psi_{\omega_2}$
>
>
> > In equation (7), how is B defined? i. How to choose the number of dimensions P, is it the same around any latent z? ii. How to choose columns in B? iii. Why a constant projection B can be used for all neighbors around the specific z?
>
> $\mathbf{B}$ is a matrix of size $P \times E$ where the rows are orthonormal s.t. $\mathbf{B}\mathbf{B}^T = \mathbb{I}_P$.
>
> i) Yes, we consider the same $\mathbf{B}$ for all latent vectors $\mathbf{z}$ during local complexity computation. Local complexity computation for a latent $\mathbf{z}$ requires 2*P forward passes through the network. We choose P=4 to minimize computation overhead.
>
> ii) We use jax.nn.init.orthogonal to obtain $\mathbf{B}$ with orthonormal rows.
>
> iii) We follow [10] and use a constant $\mathbf{B}$ for all $\mathbf{z}$.
>
> [10] Humayun et al., Deep networks always grok and here is why, ICML 2024
>
> > What explicitly are the input space vectors for DDPM and Stable Diffusions? Are they the inputs to the denoising UNet, or could they be features in the middle layers of the UNet?
>
> We apologize for the lack of clarity in the terminology. DDPM input space vectors are vectors from the 2D input space of the DDPM. For Stable Diffusion we only consider latent space vectors which are input to the UNet or the Decoder. We do not consider intermediate layer features of the UNet. However this is an interesting point that the author has raised and can be further explored in future work.
>
> > In section 3.1 L250, "input vectors within 0.05 units of the training data", how is 0.05 chosen and why it is a reasonable value?
>
> In Sec 3.1, the data manifold $\mathcal{M}$ is considered a union of circular sets of radius 0.05 centered on the training data points. 0.05 is chosen upon qualitative visualization of the circular sets. In Rebuttal Fig 7, the union of red circular regions is considered the training data manifold.
>
> Rebuttal Fig 7: Manifold $\mathcal{M}$ is considered a union of circular sets of radius 0.05 centered on the training data points.
> https://github.com/lukanon/rebuttal/blob/main/onmanifold_ddpm.png
>
>
> (e) In Figure 4, how are the three anchors (representing latent from 3 text prompts) labeled in the latent space and what are other points?
>
> In Figure 4 the top anchor corresponds to the “a cat” prompt, the bottom left anchor corresponds to the “a dog” prompt and the right anchor corresponds to the “a fox” prompts. In Appendix Figure 22-27 we present decoded images for latent vectors on the 2D slice visualized in Figure 4. Notice that away from the convex hull of the anchors, the decoded images contain many artifacts.
>
> -----
> ## Thanks
>
> We thank the reviewer for their detailed and constructive review which we believe has allowed us to significantly improve the paper. We kindly request the reviewer to ask any follow-up questions deemed necessary.

---

> > ### Comment · Reviewer_fW1y · 2024-12-01
> > **Follow-up**
> >
> > Thank the authors for the comprehensive explanation, I will update my score to 6. I have additional details and hope to make it clear:
> >
> > 1. Rebuttal Fig 5 is interesting, and could you explain why there are such changes in generated images concerning local scaling (ψ), rank (ν), and complexity (δ)?
> >
> > 2. In equation (7), B: (a) Does that mean B is randomly initialized? If it is, will the measurement be stable for different random initializations? (b) Why is a constant B for all trustworthy? (Could different local geometry require different B for accurate measurement?)

---

> ### Author Response · Authors · 2024-12-03
>
> We thank the reviewer for raising their score and for their follow-up questions.
>
> **1.** As shown in Rebuttal Fig. 5, increasing the local scaling level sets causes the images to transition from being modal to anti-modal. The highest local scaling level set contains images with low fidelity and lower shape variance, indicating anti-modality. From the local complexity level sets, we observe that the learned manifold is locally smoothest for digit 1 images. This aligns with the fact that digit 1 exhibits the lowest intra-class variability in the MNIST dataset, likely enabling the network to learn smoother manifolds locally. Images in higher local complexity level sets have lower fidelity, suggesting that regions of the manifold with less smoothness correspond to lower fidelity generation. These findings corroborate the results presented in Sections 3 and 4 of our paper. This is further evident in Rebuttal Fig. 6, which shows the joint distribution of mean squared error and local complexity. Higher local complexity level sets consistently contain images with higher mean squared reconstruction errors. However, for local rank, we do not observe a significant pattern, likely because the MNIST dataset is relatively noiseless and exhibits a narrow dynamic range for local rank.
>
> Rebuttal Fig 5. https://github.com/lukanon/rebuttal/blob/main/mnist_betavae_qualitative.png
>
> Rebuttal Fig 6. https://github.com/lukanon/rebuttal/blob/main/correlations.png
>
> **2a.** Yes, matrix $B$ is randomly initialized. Across different random initializations of $B$ we observe that the standard deviation of local complexity for a given latent is considerably smaller than the standard deviation across different samples. Additionally, since our analysis relies on distribution-level statistics—such as comparing the local complexity distributions of memorized versus non-memorized prompts—changing the random seed used to generate $B$ does not affect the overall results.
>
> **2b.** We use a constant $B$ for all samples since we want the rows of $B$ to represent identical latent space directions. This is important to ensure that we can compare the local complexity statistics between different samples. Instead of a constant randomized $B$, future work can consider semantic directions in the latent space as rows in $B$ [8] to measure complexity along meaningful directions on the image manifold.
>
> [8] Chen et al., Exploring Low-Dimensional Subspaces in Diffusion Models for Controllable Image Editing, ArXiv 2024

---

### Official Review · Reviewer_WPMu · 2024-11-05

**Soundness:** 3
**Presentation:** 2
**Contribution:** 3
**Rating:** 6
**Confidence:** 3

**Summary:**

This paper, "What Secrets Do Your Manifolds Hold? Understanding the Local Geometry of Generative Models", explores the role of local geometry in deep generative models and its effect on generation quality, diversity, and memorization. The authors propose three geometric descriptors—local scaling ($\psi$), rank ($v$), and complexity ($\delta$)—to characterize the latent space of generative models, focusing on models with Continuous Piecewise-Linear (CPWL) mappings. They empirically demonstrate that these descriptors are correlated with various aspects of generative performance and propose a reward model trained on these descriptors to guide sample generation in diffusion models, particularly Stable Diffusion.

**Strengths:**

**1. Interesting Indicators of Geometry**: This paper introduces an interesting use of geometric descriptors to understand and control generative models, providing new insights into latent space structure.

**2. Comprehensive Experiments**: The paper includes extensive experiments across various models, from toy models to Stable Diffusion.

**3. Practical Framework for Control**: The reward model offers a practical way to influence generation characteristics, such as diversity and aesthetic quality, by guiding sampling in the latent space based on geometry based on the trained scalar model of the geometry descriptor.

**Weaknesses:**

**1. Computational Demands**: The method relies on calculating Jacobians on each linear piece of model manifold. This may be computationally intensive for large models, limiting its practical application. Such a limitation potentially indicates the approach is less scalable. Besides, since the computation of each geometric descriptor is computationally demanding, generating data for training latent space reward models is also computationally demanding and therefore possibly not scalable.

**2. Descriptor Interpretation**: Some descriptors, particularly local complexity ($\delta$), lack intuitive interpretation when applied to high-dimensional latent spaces, which could be further clarified. This makes the paper more like a pure empirical computation of each descriptor instead of an in-depth study of the latent geometry of generative models.

**3. Weak Evaluations**: A major drawback of this paper is the weak lack of quantitative evaluations. For instance, the paper should provide quantitative metrics such as FIDs or Human preference scores to evaluate the performances of the proposed geometry descriptor guided sampling instead of merely qualitative plots.

**4. Poor Compatibility for Networks with Smooth Activation Functions**: If I do not misunderstand, I think the approach is only properly defined for models with piece-wise linear neural networks instead of those networks using smooth activation functions such as SiLU, GELU or SwiGLU. This might limit the broader usage of the proposed approach and the impacts of the study.

**Questions:**

**1.** I am curious about the computational costs such as GPU hours per 1k samples to calculate each descriptor value, as well as the data preparation of reward models. I think this would give readers a more comprehensive understanding of the empirical impact of the proposed approach.

**2.** I think it would be good if authors could provide more studies on DiT or MM-DiT-based text-to-image diffusion models. The UNet-based Stable Diffusion 1.4 model is kind of weak when compared with current strong DiT-based models such as Stable Diffusion 3 or Flux. This study will give readers an in-depth understanding of how compatible the approach is across different neural network architectures.

**3.** It would be good if authors could provide more intuitive discussions on each of the three descriptors. This will help readers understand the intuitions behind them.

**4.** The author should provide quantitative metrics such as FID or human preference scores when using reward guidance sampling.

---

> ### Author Response · Authors · 2024-11-26
> **Author Response 1**
>
> We would first like to thank the reviewer for their detailed review of our work and for their positive comments. We are delighted to see that the reviewer finds our analysis on geometric descriptors insightful, our experiments comprehensive, and our geometry based reward modeling framework a practical method to control generation. We answer specific comments/questions by the reviewer below:
>
> --------------------------------
>
> ## Quantitative evaluations of the reward model
> > “The author should provide quantitative metrics such as FID or human preference scores when using reward guidance sampling.”
>
> We agree with the reviewer and provide additional experiments to quantitatively evaluate the performance of our local scaling based reward model.
>
> **Data preparation.** We obtain training data for the reward model by i) sampling $N$ images from Imagenet and encoding them to the Stable Diffusion latent space ii) adding noise using the forward diffusion process up to randomly chosen noise levels iii) for each latent computing the local scaling descriptor.
>
> To evaluate the performance of the reward model and dependency of the reward model on the number of training samples, we train multiple models for $N=${ $50K, 200K, 400K, 800K$ }. For evaluation, we generate $2560$ samples using the dreambooth live subject prompt templates [1], with Imagewoof [2] dogs as subjects that Stable Diffusion can generate with good variety. While Imagewoof dog classes are present in Imagenet therefore possibly in the training data, the dreambooth prompt templates contain a variety of settings that are not generally present in Imagenet, e.g., `a <subject> on top of pink fabric`.
>
> **Evaluation Setup.** We first sample Stable diffusion without any reward guidance and with classifier-free guidance of 7.5 to obtain baseline samples. We partition the range of local scaling values obtained for the baseline samples into $n=10$ bins (Rebuttal Fig 1 x-axis), where each bin contains images from a local scaling level set. Following that we use the same seed and prompts as the baseline samples to generate images using reward guidance to increase local scaling. For each bin or pre-guidance local scaling level set, we compare between the baseline samples and corresponding reward guided generations in the following three axes: i) change of local scaling ii) change of vendi (diversity) score iii) the change of human preference score (RAHF [3] aesthetic score).
>
> **Results.** In Rebuttal Fig.1-left below (anonymized link), we present the mean local scaling per bin with 95% confidence interval. Here the blue line represents the mean pre-guidance local scaling values, increasing from left to right. In Rebuttal Fig 1-middle and Rebuttal Fig-1 right, we present vendi scores and average predicted human preference scores [3]. For any bin, we present results for the reward guidance scale that maximizes the local scaling. We see that even for a model trained with $50K$ samples, we can have a considerable increase in the local scaling, diversity and aesthetic score for most of the bins. Changes in local scaling, vendi and aesthetic scores are higher for the lower pre-guidance local scaling level set bins compared to the higher pre-guidance local scaling level set bins.
>
> Rebuttal Fig 1. https://github.com/lukanon/rebuttal/blob/main/reward_guidance_effect.png
>
> We are adding the quantitative evaluation results to section 5 and will provide the updated manuscript shortly.
>
> [1] Ruiz et. al, DreamBooth: Fine Tuning Text-to-Image Diffusion Models for Subject-Driven Generation, CVPR 23.
>
> [2] https://github.com/fastai/imagenette
>
> [3] Liang et. al, Rich Human Feedback for Text-to-Image Generation, CVPR 24

---

> ### Author Response · Authors · 2024-11-26
> **Author Response 2**
>
> --------------------------------
>
> ## Computational complexity due to number of regions and jacobian computation
>
> > ” The method relies on calculating Jacobians on each linear piece of model manifold. This may be computationally intensive for large models, limiting its practical application. Such a limitation potentially indicates the approach is less scalable. Besides, since the computation of each geometric descriptor is computationally demanding, generating data for training latent space reward models is also computationally demanding and therefore possibly not scalable.”
>
> While our analysis stems from the continuous piece-wise linear (CPWL) assumption of neural networks, it does not require us to compute the jacobians for every linear region in the CPWL partition. We agree with the reviewer that such a requirement would have combinatorial complexity and for larger networks the jacobian computation would be expensive. As we have presented above, in practice we compute local scaling for a smaller and finite number of images from Imagenet to train our reward models. Even with only $50K$ training samples, we observe a considerable increase in local scaling via reward guidance as was discussed in the previous response and Rebuttal Fig 1.
>
> To avoid computing the singular values using the full input-output jacobian – which will be significantly expensive for large networks – for local scaling/rank computation we obtain singular values via randomized SVD [4]. First we obtain a random projection matrix with orthonormal rows $\mathbf{W}$ with shape $k \times n$ such that $\mathbf{W}\mathbf{W}^T = \mathbb{I}_k$. Here $n$ is the dimensionality of the outputs generated by the network. We therefore approximate local scaling as:
>
> $\psi_\omega^{(trunc)} = \sum_{i=1}^k log(\sigma_i^{(trunc)})$, where $\sigma_i^{(trunc)}$ are the non-zero singular values of $\mathbf{W}\mathbf{A}_\omega$.
>
> For any $\omega$, if $\mathbf{W}$ forms a basis for the range of $\mathbf{A_\omega}$ then $\sigma_i  \approx \sigma_i^{(trunc)} \forall i={1,2…k}$ [4]. Therefore $\mathbf{WA_\omega}$ would provide us a low-rank approximation of $\mathbf{A_\omega}$.
>
> In our experiments we have tried two methods to obtain the projection matrix $W$
> 1) by obtaining the eigenvectors for the covariance matrix for a set of 50K randomly generated samples. This was suggested in [4].
> 2) by performing QR decomposition of a randomly initialized matrix. Implemented in jax.nn.initializers.orthogonal [5]
>
> We see that the performance difference between methods 1) and 2) are negligible therefore consider the cheaper alternative 2) and consider a fixed pre-computed $\mathbf{W}$ with k=120 for all $\mathbf{A_\omega}$ in our Stable Diffusion experiments.
>
> > “I am curious about the computational costs such as GPU hours per 1k samples to calculate each descriptor value, as well as the data preparation of reward models. I think this would give readers a more comprehensive understanding of the empirical impact of the proposed approach.”
>
> The computation times for the local scaling and local rank computation (since both require one randomized SVD computation for one latent vector) ends up being 3929s for 1000 samples. For local complexity we require 113s for 1000 samples. All the estimates are for a JAX implementation of Stable Diffusion on TPUv3.
>
> Note that to train a reward model, we require the descriptors to be computed only once for each pre-trained model. If we compute the local scaling for 100k samples we require 173.1 TPU v3 hours which is equivalent to 54.58 V100 hours (according to Appendix A.3 [6]). Compared to ~79,000 A100 hours required for Stable Diffusion training [7], ~24000 hours with enterprise level optimization [8], the computation required for the descriptors and reward model training is significantly small. The computation time for the local descriptors can be further reduced by using a smaller $k$ for our projection matrix $W$, or by using non-jacobian based methods, e.g., estimating the local scaling by measuring the change of volume for a unit norm $\ell_1$-ball at the input. We leave exploration of these directions for future work.
>
> We will expand Section 2.2 *“Extending Beyond Continuous Piecewise-Linear Generators”* with all the aforementioned details and add implementation pseudocode in the appendix. We thank the reviewer once again for the comment, we believe this is an important point that requires being highlighted in the paper to improve clarity and readability.
>
> [4] Halko et. al, Finding Structure with Randomness: Probabilistic Algorithms for Constructing Approximate Matrix Decompositions, SIAM Review 2011.
>
> [5] https://jax.readthedocs.io/en/latest/_autosummary/jax.nn.initializers.orthogonal.html
>
> [6] Dhariwal et. al, Diffusion Models Beat GANs on Image Synthesis
>
> [7] https://www.mosaicml.com/blog/training-stable-diffusion-from-scratch-costs-160k
>
> [8] https://www.databricks.com/blog/stable-diffusion-2

---

> ### Author Response · Authors · 2024-11-26
> **Author Response 3**
>
> ------------------
>
> ## Adding clarity and more intuition for descriptors
>
> > “Some descriptors, particularly local complexity ($\delta$), lack intuitive interpretation when applied to high-dimensional latent spaces, which could be further clarified”. “It would be good if authors could provide more intuitive discussions on each of the three descriptors. This will help readers understand the intuitions behind them.”
>
> We apologize for the lack of clarity in the local descriptor introduction/discussions in Sec. 2. Neural networks with only CPWL non-linearities are CPWL operators, and their input-output mapping can be fully characterized by the regions in the input space partitioning and the affine operator per region (see Equation 1). This is irrespective of the dimensionality of the input space or latent space [9]. Given that, we can measure three characteristics of the CPWL mapping via the three descriptors. **Local scaling** characterizes the change of volume by the affine slope $\mathbf{A_\omega}$ going from the latent space to the data manifold. **Local rank** characterizes the number of dimensions retained on the manifold after the network locally scales the latent space. Both local rank and scaling quantify first order properties of the CPWL operator. Local complexity approximates the ‘number of unique affine maps’ within a given neighborhood [10] by computing the number of CPWL knots intersecting an $\ell_1$ ball in the input/latent space. Therefore local complexity is a measure of ‘un-smoothness’ and quantifies local second-order properties of a CPWL operator.
>
> **Correlations between local scaling and rank.** By definition, local scaling and local rank are correlated, since both characterize the change of volume by the network input-output map at any input space linear region – also evident in Equations 2 and 4. This is also evident for our low dimensional DDPM setting presented in Fig 3., local rank and local scaling are highly correlated. The correlation persists throughout training as can be seen in Fig.3 right panel top and bottom. However in Fig 4., we can see that in the high-dimensional Stable Diffusion latent space, local scaling and rank are correlated but local rank has sharper changes spatially compared to local scaling.
>
> **Correlations between local complexity and rank.** There also exist correlations between local complexity and local rank due to the continuity of CPWL maps – between two neighboring linear regions $\omega_1$ and $\omega_2$, the corresponding slope matrices $\mathbf{A_{\omega_1}}$ and $\mathbf{A_{\omega_2}}$ differ by at most one row. Therefore between two neighboring regions the difference in local rank can be at most one.  Informally, the local rank in a neighborhood $V$ is lower bounded by the number of non-linearities in neighborhood $V$. This is evident in the empirical results presented in Fig 3 and Fig 4. In both figures, for input space neighborhoods with higher local complexity, we see a decrease in local rank. However, we do not observe sharp changes in local complexity as we observe in local rank in Fig 4. In Fig 3 we see that local rank is more discriminative of the data manifold compared to local complexity. Their training and denoising dynamics differ significantly as seen in Fig 3 rightmost column.
>
> **Local complexity in high dimensional spaces.** With $E$ as the latent space dimensionality and $M$ as the number of neurons in the network, the local complexity for any network is $\mathcal{O}(M^E)$. While prior work has provided bounds for local complexity in high dimensional spaces [11], the study of local complexity in high dimensional spaces require more empirical exploration than theoretical for large scale deep neural networks. That is why in Sec 3 and Sec 4 of our paper, we perform a number of exploratory experiments to establish intuition for the local descriptors.
>
>
> **Qualitative and quantitative results on correlations.** We train a beta-VAE unconditionally on MNIST and present in Rebuttal Fig 5 samples from increasing local descriptor level sets from left to right along the columns. In rebuttal Fig 6, we present joint distributions of local scaling, complexity, rank and mean squared reconstruction error for training and test samples. We see that while local scaling, complexity and rank have some linear correlation, the classwise distribution is very different between the three (Rebuttal Fig 5).
>
> Rebuttal Fig 5. https://github.com/lukanon/rebuttal/blob/main/mnist_betavae_qualitative.png
>
> Rebuttal Fig 6. https://github.com/lukanon/rebuttal/blob/main/correlations.png
>
> [9] Balestriero et. al, A Spline Theory of Deep Learning, ICML 2018
>
> [10] Humayun et. al, Deep networks always grok and here is why, ICML 2024
>
> [11] Hanin et. al, Complexity of Linear Regions in Deep Networks, ICML 2019

---

> ### Author Response · Authors · 2024-11-26
> **Author Response 4**
>
> -----------------
>
> ## Compatibility for networks with smooth activations
>
> >  "If I do not misunderstand, I think the approach is only properly defined for models with piece-wise linear neural networks instead of those networks using smooth activation functions such as SiLU, GELU or SwiGLU. This might limit the broader usage of the proposed approach and the impacts of the study."
>
> We thank the reviewer for the clarifying question. While the descriptors are defined for CPWL mappings, modern generative models employ a mixture of CPWL and non-CPWL operations. For networks with smooth activation functions or non-piecewise-linear non-linearities, our descriptors equate first order Taylor approximations. We agree with the reviewer that there may be approximation errors incurred when we move from ReLU to smooth variants. However, Stable Diffusion already employs the GeLU activation function for which we perform the bulk of our experiments and find strong connections between the approximate local geometry and downstream generation. This is because smooth activation functions induce a soft VQ partitioning of the latent space compared to the hard VQ partitioning induced by a CPWL map [12]. This suggest that much of the local linear structure we expect in CPWL maps are retained even if we employ smooth approximations of ReLU. Recent work has also empirically verified the local linearity for a large class of image based diffusion models [13].
>
> [12]  Balestriero et al., From Hard to Soft: Understanding Deep Network Nonlinearities via Vector Quantization and Statistical Inference, ICLR 2019
>
> [13] Chen et. al, Exploring Low-Dimensional Subspaces in Diffusion Models for Controllable Image Editing, NeurIPS 2024
>
> -----------------
>
> ## Experiments on newer DiT-based models
>
> > "I think it would be good if authors could provide more studies on DiT or MM-DiT-based text-to-image diffusion models."
>
> We thank the reviewer for raising this important point. Since we are based on the CPWL formulation of NNs, our framework would generalize to models of any scale and any architecture with CPWL non-linearities. Empirically we have shown it to generalize for non-CPWL architectures like Stable Diffusion v1.4 and DDPM that employs non CPWL non-linearities such as attention, GeLU and much more. Following suggestions by the reviewer, we have performed additional experiments with a DiT-XL [7] trained on Imagenet-256. For the DiT we compute the descriptors for the transformer network, conditioned on noise level $t=0$, i.e., zero noise level. We generate 5120 images conditioned on Imagewoof [2] classes and present in Rebuttal Fig 2 below, increasing local scaling level sets from left to right. We see that similar to Fig 6 from the paper, DiT exhibits a qualitative correlation between visual complexity and local scaling. For additional analysis we repeat the Stable Diffusion experiments from Sec 4 (Figure 9) on the relation between diversity and local scaling for DiT. We see that similar to Stable Diffusion, for increasing local scaling level sets, the diversity of images increase and then drop for the highest local scaling level sets.
>
> Rebuttal Fig 2: Randomly sampled images from local scaling level sets for which Vendi scores are presented in Fig 2
> https://github.com/lukanon/rebuttal/blob/main/imagewoof_combined_levelsets.jpg
>
> Rebuttal Fig 3: Vendi score for increasing local scaling level sets, computed for a DiT-XL Transformer network trained on Imagenet.
> https://github.com/lukanon/rebuttal/blob/main/imagewoof_combined_vendi.pdf
>
> [7]  Peebles et. al, Scalable Diffusion Models with Transformers, ICCV 23, URL: https://github.com/facebookresearch/DiT
>
>
> ----------
>
> ## Thanks
>
> We thank the reviewer for their detailed and constructive review which we believe has allowed us to significantly improve the paper. We kindly request the reviewer to ask any follow-up questions deemed necessary. We will update the manuscript shortly with the suggested edits.

---

> ### Comment · Reviewer_WPMu · 2024-12-02
> **Thanks to authors rebuttal.**
>
> I appreciate the authors' rebuttal, which has completely resolved my concerns.
>
> Though I think the paper is more of an intuitive study with a large remaining space to explore, I would like to raise my score to 6 to encourage authors to future explore numerically efficient geometric descriptors, and potentially more solid empirical algorithms.

---

> > ### Author Response · Authors · 2024-12-03
> >
> > We sincerely thank the reviewer for their thoughtful feedback and for raising the score. The reviewer's insights have been instrumental in improving the quality of the paper, refining the implementation details, and enhancing the discussion of the generalizability of the proposed geometric descriptors. In our work, we introduce geometric descriptors, provide intuition on their relationship with downstream generation, and demonstrate their potential applications, e.g., geometry based guidance to enhance diversity in generated samples.
> >
> > We fully agree with the reviewer that there are numerous promising future directions to explore. For instance, the geometric descriptors could be employed to constrain reward-based generation on the manifold, thereby mitigating reward hacking. Similarly, they could enable descriptor-based regularization during model adaptation or unlearning processes. We believe that since the descriptors are general and model agnostic, our work lays the groundwork for further advancements in leveraging the "learned geometry" of generative models to improve downstream performance.

---

### Author Response · Authors · 2024-12-04
**Summary of Author-Reviewer Discussions**

We thank the reviewers for their detailed and constructive review of our work. Here we provide a summary of our paper, the reviewer comments and rebuttals.

-----
## Paper Summary
-----
In this paper we study the local geometry of generative models and their relationship with downstream generation. We propose using three descriptors to characterize the local geometry– local scaling $(\psi)$ which quantifies the local change of volume by a generator, local rank $(\nu)$ which quantifies the local dimensionality of the output manifold and local complexity $(\delta)$ which quantifies the local smoothness of the generator input-output mapping. Through extensive experiments for DDPMs trained on 2D datasets, beta-VAE trained on MNIST, DiT-XL trained on Imagenet (added during rebuttal), Latent Diffusion Model trained on CelebAHQ (added during rebuttal) and Stable Diffusion trained on LAION, we find that i) local scaling, rank and complexity of a pre-trained model is sensitive to the training distribution and can discriminate between on and off manifold neighborhoods ii) local scaling and rank can discriminate prompts memorized by Stable Diffusion iii) local scaling is correlated with downstream visual complexity and generation diversity iv) local scaling is correlated with generation aesthetics. We also demonstrate qualitatively and quantitatively (added during rebuttal) that guiding generation based on local scaling can increase diversity and human preference for Stable Diffusion generated images.

---
## Reviewer comments
---

All the reviewers have found the experiments comprehensive, e.g., “the authors conduct extensive experiments in multiple scenarios” by reviewer KSLy, “I believe this is a very strong empirical exploration paper” by reviewer dk5T, “studying the local geometry from a comprehensive perspective” by reviewer fW1y, “comprehensive experiments” and “extensive experiments across various models, from toy models to Stable Diffusion” by reviewer WPMu. The reviewers have also positively commented on the novelty and impact of the paper “this paper will significantly impact the study of image manifold geometry” and ”intuitive and well-motivated” by reviewer dk5T, “the idea is novel” by reviewer KSLy.

-----
## Reviewer concerns and rebuttal
-----


**a) Quantitative evaluation of the geometry based reward model. Reviewers WPMu, fW1y, KSLy, dk5T**

We have provided new quantitative evaluations of reward models trained on local scaling with increasing training data volume. We provide quantitative evidence of increased diversity and increased human preference score with geometry reward guidance. [(Link)](https://openreview.net/forum?id=etif9j1CnG&noteId=U3U5QLSI5J)


**b) Implementation details and computational complexity. Reviewers WPMu, fW1y, KSLy, dk5T**

Reviewers had asked to clarify the complexity of computing the input-output jacobian of networks, required for local scaling and rank computation. We have clarified that we use randomized SVD to significantly reduce compute requirements and have provided implementation details. We have also provided computation times required for a JAX implementation of Stable Diffusion on TPUv3. [(Link)](https://openreview.net/forum?id=etif9j1CnG&noteId=fumE137MkM)

**c) Correlations/redundancy between the three descriptors. Reviewers WPMu, fW1y, KSLy, dk5T**

We have clarified that while the three descriptors are correlated by definition, they quantify distinct geometric quantities therefore are not redundant. While in experiments we do observe correlations between the descriptors, there are counter examples that we have highlighted in our rebuttals [(Link)](https://openreview.net/forum?id=etif9j1CnG&noteId=tcytg0anRl).


**d) Compatibility with non-continuous piecewise linear networks. Reviewers WPMu, fW1y**

Reviewers asked for justification of the continuous piecewise linear (CPWL) assumption and whether it works for non-piecewise linear functions. We have clarified that the CPWL is not a strict requirement for results to hold, and most of our experiments are performed on non-CPWL networks [(Link)](https://openreview.net/forum?id=etif9j1CnG&noteId=7HYrUZoMJF)

**e) Applicability to newer models. Reviewer WPMu**

We have provided additional results for a transformer based diffusion model DiT-XL [(Link)](https://openreview.net/forum?id=etif9j1CnG&noteId=0PCqBUP0hZ)

**f) Literature review and related works fW1y, dk5T**

We have performed a review of related literature and provided discussions comparing concurrent work with ours. [(Link)](https://openreview.net/forum?id=etif9j1CnG&noteId=njbuqIipvt)

We are also improving the overall writing to increase the clarity of our paper. We have brought a number of changes in the manuscript based on the rebuttals, e.g., added literature review to appendix, implementation details in appendix and Sec 2.2, moved Fig 3 and Fig 5 to appendix, and added quantitative evaluation results for the reward model in Sec 5.

---

### Meta-Review · Area_Chair_mhLV · 2024-12-17

**Metareview:**

This paper explores the local geometry of generative models and its impact on generation quality. Using the theory of continuous piecewise-linear generators, the authors introduce three metrics: (1) local rank, (2) local scaling, and (3) local complexity. They demonstrate that local scaling correlates with scene complexity and can detect whether data lies on the input (training) manifold. Additionally, they train a reward model to approximate local scaling, showing that increasing its value leads to more objects and complex backgrounds in generated outputs.

All reviewers agree that this work is interesting and above the acceptance bar.

**Additional Comments On Reviewer Discussion:**

During the rebuttal, the authors successfully addressed most reviewers' concerns with more comprehensive evaluations and improved writing with better clarity.

---

### Decision · Program_Chairs · 2025-01-22

Accept (Poster)